# Distinct temporal difference error signals in dopamine axons in three regions of the striatum in a decision-making task

Iku Tsutsui-Kimura[1], Hideyuki Matsumoto[1,2], Korleki Akiti[1], Melissa M Yamada[1], Naoshige Uchida[1], Mitsuko Watabe-Uchida[1]*

[1]Department of Molecular and Cellular Biology, Center for Brain Science, Harvard University, Cambridge, United States; [2]Department of Physiology, Osaka City University Graduate School of Medicine, Osaka, Japan

**Abstract** Different regions of the striatum regulate different types of behavior. However, how dopamine signals differ across striatal regions and how dopamine regulates different behaviors remain unclear. Here, we compared dopamine axon activity in the ventral, dorsomedial, and dorsolateral striatum, while mice performed a perceptual and value-based decision task. Surprisingly, dopamine axon activity was similar across all three areas. At a glance, the activity multiplexed different variables such as stimulus-associated values, confidence, and reward feedback at different phases of the task. Our modeling demonstrates, however, that these modulations can be inclusively explained by moment-by-moment *changes* in the expected reward, that is the temporal difference error. A major difference between areas was the overall activity level of reward responses: reward responses in dorsolateral striatum were positively shifted, lacking inhibitory responses to negative prediction errors. The differences in dopamine signals put specific constraints on the properties of behaviors controlled by dopamine in these regions.

*For correspondence: mitsuko@mcb.harvard.edu

## Introduction

Flexibility in behavior relies critically on an animal's ability to alter its choices based on past experiences. In particular, the behavior of an animal is greatly shaped by the consequences of specific actions – whether a previous action led to positive or negative experiences. One of the fundamental questions in neuroscience is how animals learn from rewards and punishments.

A neurotransmitter dopamine, is thought to be a key regulator of learning from rewards and punishments (*Hart et al., 2014*; *Montague et al., 1996*; *Schultz et al., 1997*). Neurons that release dopamine (hereafter, dopamine neurons) are located mainly in the ventral tegmental area (VTA) and substantia nigra pars compacta (SNc). These neurons send their axons to various regions including the striatum, neocortex, and amygdala (*Menegas et al., 2015*; *Yetnikoff et al., 2014*). The striatum, which receives the densest projection from VTA and SNc dopamine neurons, is thought to play particularly important roles in learning from rewards and punishments (*Lloyd and Dayan, 2016*; *O'Doherty et al., 2004*). However, what information dopamine neurons convey to the striatum, and how dopamine regulates behavior through its projections to the striatum remains elusive.

A large body of experimental and theoretical studies have suggested that dopamine neurons signal reward prediction errors (RPEs) – the discrepancy between actual and predicted rewards (*Bayer and Glimcher, 2005*; *Cohen et al., 2012*; *Hart et al., 2014*; *Schultz et al., 1997*). In particular, the activity of dopamine neurons resembles a specific type of prediction error, called temporal difference error (TD error) (*Montague et al., 1996*; *Schultz et al., 1997*; *Sutton, 1988*; *Sutton, 1987*). Although it was widely assumed that dopamine neurons broadcast homogeneous RPEs to a swath of dopamine-recipient areas, recent findings indicated that dopamine signals are more

diverse than previously thought (*Brown et al., 2011*; *Kim et al., 2015*; *Matsumoto and Hikosaka, 2009*; *Menegas et al., 2017*; *Menegas et al., 2018*; *Parker et al., 2016*). For one, recent studies have demonstrated that a transient ('phasic') activation of dopamine neurons occurs near the onset of a large movement (e.g. locomotion), regardless of whether these movements are immediately followed by a reward (*Howe and Dombeck, 2016*; *da Silva et al., 2018*). These phasic activations at movement onsets have been observed in somatic spiking activity in the SNc (*da Silva et al., 2018*) as well as in axonal activity in the dorsal striatum (*Howe and Dombeck, 2016*). Another study showed that dopamine axons in the dorsomedial striatum (DMS) are activated when the animal makes a contralateral orienting movement in a decision-making task (*Parker et al., 2016*). Other studies have also found that dopamine axons in the posterior or ventromedial parts of the striatum are activated by aversive or threat-related stimuli (*de Jong et al., 2019*; *Menegas et al., 2017*). An emerging view is that dopamine neurons projecting to different parts of the striatum convey distinct signals and support different functions (*Cox and Witten, 2019*).

Previous studies have shown that different parts of the striatum control distinct types of reward-oriented behaviors (*Dayan and Berridge, 2014*; *Graybiel, 2008*; *Malvaez and Wassum, 2018*; *Rangel et al., 2008*). First, the ventral striatum (VS) has often been associated with Pavlovian behaviors, where the expectation of reward triggers relatively pre-programmed behaviors (approaching, consummatory behaviors etc.) (*Dayan and Berridge, 2014*). Psychological studies suggest that these behaviors are driven by stimulus-outcome associations (*Kamin, 1969*; *Pearce and Hall, 1980*; *Rescorla and Wagner, 1972*). Consistent with this idea, previous experiments have shown that dopamine in VS conveys canonical RPE signals (*Menegas et al., 2017*; *Parker et al., 2016*), and support learning of values associated with specific stimuli (*Clark et al., 2012*). In contrast, the dorsal part of the striatum has been linked to instrumental behaviors, where animals acquire an arbitrary action that leads to a reward (*Montague et al., 1996*; *Suri and Schultz, 1999*). Instrumental behaviors are further divided into two distinct types: goal-directed and habit (*Dickinson and Weiskrantz, 1985*). Goal-directed behaviors are 'flexible' reward-oriented behaviors that are sensitive to a causal relationship ('contingency') between action and outcome, and can quickly adapt to changes in the value of the outcome (*Balleine and Dickinson, 1998*). After repetition of a goal-directed behavior, the behavior can become a habit which is characterized by insensitivity to changes in the outcome value (e.g. devaluation) (*Balleine and O'Doherty, 2010*). According to psychological theories, goal-directed and habitual behaviors are supported by distinct internal representations: action-outcome and stimulus-response associations, respectively (*Balleine and O'Doherty, 2010*). Lesion studies have indicated that goal-directed behaviors and habit are controlled by DMS and the dorsolateral striatum (DLS), respectively (*Yin et al., 2004*; *Yin et al., 2005*).

Instrumental behaviors are shaped by reward, and it is generally thought that dopamine is involved in their acquisition (*Gerfen and Surmeier, 2011*; *Montague et al., 1996*; *Schultz et al., 1997*). However, how dopamine is involved in distinct types of instrumental behaviors remains unknown. A prevailing view in the field is that habit is controlled by 'model-free' reinforcement learning, while goal-directed behaviors are controlled by 'model-based' mechanisms (*Daw et al., 2005*; *Dolan and Dayan, 2013*; *Rangel et al., 2008*). In this framework, habitual behaviors are driven by 'cached' values associated with specific actions (action values) which animals learn through direct experiences via dopamine RPEs. In contrast, goal-directed behaviors are controlled by a 'model-based' mechanism whereby action values are computed by mentally simulating which sequence of actions lead to which outcome using a relatively abstract representation (model) of the world. Model-based behaviors are more flexible compared to model-free behaviors because a model-based mental simulation may allow the animal to compute values in novel or changing circumstances. Although these ideas account for the relative inflexibility of habit over model-based, goal-directed behaviors, they do not necessarily explain the most fundamental property of habit, that is, its insensitivity to changes in outcome, as cached values can still be sensitive to RPEs when the actual outcome violates expectation, posing a fundamental limit in this framework (*Dezfouli and Balleine, 2012*; *Miller et al., 2019*). Furthermore, the idea that habits are supported by action value representations does not necessarily match with the long-held view of habit based on stimulus-response associations.

Until recently an implicit assumption across many studies was that dopamine neurons broadcast the same teaching signals throughout the striatum to support different kinds of learning (*Rangel et al., 2008*; *Samejima and Doya, 2007*). However, as mentioned before, more recent

studies revealed different dopamine signals across striatal regions, raising the possibility that different striatal regions receive distinct teaching signals. In any case, few studies have directly examined the nature of dopamine signals across striatal regions in instrumental behaviors, in particular, between DLS and other regions. As a result, it remains unclear whether different striatal regions receive distinct dopamine signals during instrumental behaviors. Are dopamine signals in particular areas dominated by movement-related signals? Are dopamine signals in these areas still consistent with RPEs or are they fundamentally distinct? How are they different? Characterizing dopamine signals in different regions is a critical step toward understanding how dopamine may regulate distinct types of behavior.

In the present study, we sought to characterize dopamine signals in different striatal regions (VS, DMS and DLS) during instrumental behaviors. We used a task involving both perceptual and value-based decisions in freely-moving mice – a task that is similar to those previously used to probe various important variables in the brain such as values, biases (*Rorie et al., 2010*; *Wang et al., 2013*), confidence (*Hirokawa et al., 2019*; *Kepecs et al., 2008*), belief states (*Lak et al., 2017*), and response vigor (*Wang et al., 2013*). In this task, the animal goes through various movements and mental processes – self-initiating a trial, collecting sensory evidence, integrating the sensory evidence with reward information, making a decision, initiating a choice movement, committing to an option and waiting for reward, receiving an outcome of reward or no reward, and adjusting internal representations for future performance using RPEs and confidence. Compared to Pavlovian tasks, which have been more commonly used to examine dopamine RPEs, the present task has various factors with which to contrast dopamine signals between different areas.

Contrary to our initial hypothesis, dopamine signals in all three areas showed similar dynamics, going up and down in a manner consistent with TD errors, reflecting moment-by-moment *changes* in the expected future reward (i.e. state values). Notably, although we observed correlates of accuracy and confidence in dopamine signals, consistent with previous studies (*Engelhard et al., 2019*; *Lak et al., 2017*), the appearance of these variables was timing- and trial type-specific. In stark contrast with these previous proposals, our modeling demonstrates that these apparently diverse dopamine signals can be inclusively explained by a single variable – TD error, that is moment-by-moment changes in the expected reward in each trial. In addition, we found consistent differences between these areas. For instance, DMS dopamine signals were modulated by contralateral orienting movements, as reported previously (*Parker et al., 2016*). Furthermore, DLS dopamine signals, while following TD error dynamics, were overall more positive, compared to other regions. Based on these findings, we present novel models of how these distinct dopamine signals may give rise to distinct types of behavior such as flexible versus habitual behaviors.

## Results

### A perceptual decision-making task with reward amount manipulations

Mice were first trained in a perceptual decision-making task using olfactory stimuli (*Figure 1*; *Uchida and Mainen, 2003*). To vary the difficulty of discrimination, we used two odorants mixed with different ratios (*Figure 1A*). Mice were required to initiate a trial by poking their nose into the central odor port, which triggered a delivery of an odor mixture. Mice were then required to move to the left or right water port depending on which odor was dominant in the presented mixture. Odor-water side (left or right) rule was held constant throughout training and recording in each animal. In order to minimize temporal overlaps between different trial events and underlying brain processes, we introduced a minimum time required to stay in the odor port (for 1 s) and in the water port (for 1 s) to receive a water reward.

After mice learned the task, the water amounts at the left and right water ports were manipulated (*Lak et al., 2017*; *Rorie et al., 2010*; *Wang et al., 2013*) in a probabilistic manner. In our task, one of the reward ports was associated with a big or medium size of water (BIG side) while another side was associated with a small or medium size of water (SMALL side) (*Figure 1A*). In a daily session, there were two blocks of trials, the first with equal-sized water and the second with different distributions of water sizes on the two sides (BIG versus SMALL side). The reward ports for BIG or SMALL conditions stayed unchanged within a session and were randomly chosen for each session. In each reward port (BIG or SMALL side), which of the two reward sizes was delivered was randomly

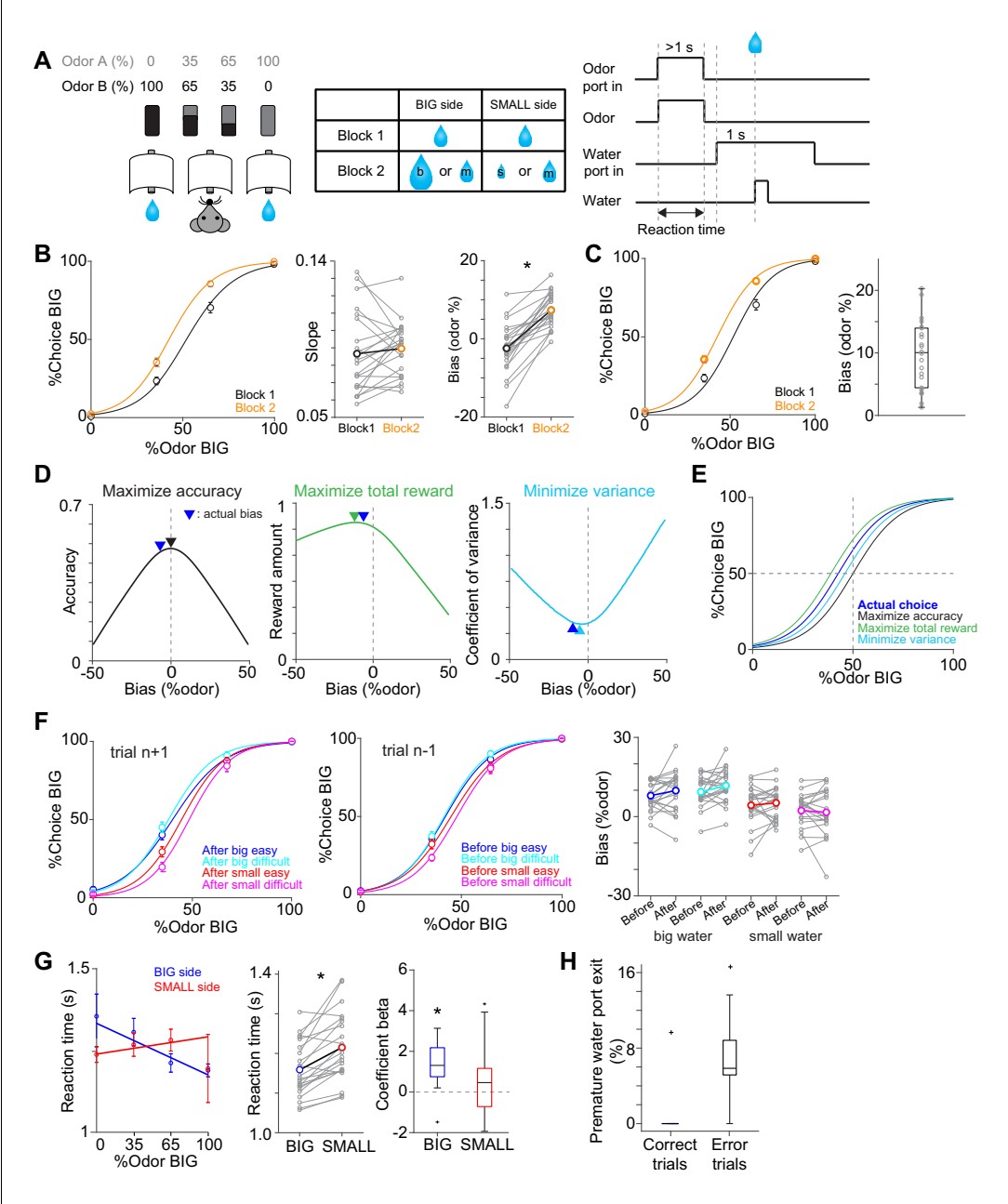

**Figure 1.** Perceptual choice paradigm with probabilistic reward conditions. (A) A mouse discriminated a dominant odor in odor mixtures that indicates water availability in either the left or right water port. Correct choice was rewarded by a drop of water. In each session, an equal amount of water was assigned at both water ports in the first block, and in the second block, big/medium water (50%–50%, randomized) was assigned at one water port (BIG side) and medium/small water (50%–50%, randomized) was assigned at another port (SMALL side). The BIG or SMALL side was assigned to a left or right water port in a pseudorandom order across sessions. (B) Left, % of choice of the BIG side in block 1 and 2 (mean ± SEM) and the average psychometric curve for each block. Center, slope of the psychometric curve was not different between blocks (t(21) = 0.75, p=0.45, paired t-test). Right, choice bias at 50/50 choice, expressed as 50 - odor (%). Choice biased toward BIG side in block 2 (t(21) = 8.5, p=2.8 × 10$^{-8}$, paired t-test). (C) Left, % of choice of the BIG side in block 1 and 2 (mean ± SEM) and the average psychometric curve with a fixed slope across blocks. Right, all the animals showed choice bias toward BIG side in block two compared to block 1 (z = 4.1, p=4.0 × 10$^{-5}$, Wilcoxon signed rank test). The choice bias was expressed by a lateral shift of a psychometric curve with a fixed slope across blocks. (D) Average reward amounts, accuracy, and coefficients of variance were examined with different levels of choice bias with a fixed slope (average slope of all animals). (E) Optimal choice patterns with different strategies in D (bias −11, 0, and −4, respectively) and the actual average choice pattern (mean bias −7.3). (F) Trial-by-trial choice updating was examined by comparing choice bias before (center, trial n−1) and after (left, trial n+1) specific trial types. Choice updating in one trial was not significant for reward acquisition of either small or big water in easy or difficult trials (right, big easy, z = −1.1, p=0.24; big difficult, z = −1.6, p=0.10; small easy, z = −0.95,

*Figure 1 continued on next page*

*Figure 1 continued*

p=0.33; small difficult, z = 0.081, p=0.93, Wilcoxon signed rank test). (G) Left, animal's reaction time was modulated by odor types. Center, for easy trials (pure odors, correct choice), reaction time was shorter when animals chose the BIG side (t(21) = −5.0, p=4.9 × 10⁻⁵, paired t-test). Right, the reaction time was negatively correlated with sensory evidence for choice of the BIG side (t(21) = −4.7, p=1.2 × 10⁻⁴, one sample t-test), whereas the modulation was not significant for choice of the SMALL side (t(21) = −1.5, p=0.13, one sample t-test). (H) Animals showed more premature exit of water port (<1 s) in trials with error choice than trials with correct choice (t(21) = −7.9, p=9.5 × 10⁻⁸, paired t-test). n = 22 animals.

The online version of this article includes the following source data and figure supplement(s) for figure 1:

**Source data 1.** Summary statistics.
**Figure supplement 1.** Average psychometric curve in odor manipulation blocks.

assigned in each trial. Note that the medium-sized reward is delivered with the probability of 0.5 for every correct choice at either side. This design was used to facilitate our ability to characterize RPE-related responses even after mice were well trained (*Tian et al., 2016*). First, the responses to the medium-sized-reward allowed us to characterize how 'reward expectation' affects dopamine reward responses because we can examine how different levels of expectation, associated with the BIG and SMALL side, affect dopamine responses to reward of the same (medium) amount. Conversely, for a given reward port, two sizes of reward allowed us to characterize the effect of 'actual reward' on dopamine responses, by comparing the responses when the actual reward was smaller versus larger than expected.

We first characterized the choice behavior by fitting a psychometric function (a logistic function). Compared to the block with equal-sized water, the psychometric curve was shifted laterally to the BIG side (*Figure 1B*, *Figure 1—figure supplement 1*). The fitted psychometric curves were laterally shifted whereas the slopes were not significantly different across blocks (t(21) = 0.75, p=0.45, n = 22, paired t-test) (*Figure 1B*). We, therefore, quantified a choice bias as a lateral shift of the psychometric curve with a fixed slope in terms of the % mixture of odors for each mouse (*Figure 1C*; *Wang et al., 2013*). All the mice exhibited a choice bias toward the BIG side (22/22 animals). Because a 'correct' choice (i.e. whether a reward is delivered or not) was determined solely by the stimulus in this task, biasing their choices away from the 50/50 boundary inevitably lowers the choice accuracy (or equivalently, the probability of reward). For ambiguous stimuli, however, mice could go for a big reward, even sacrificing accuracy, in order to increase the long-term gain; choice bias is potentially beneficial if taking a small chance of big reward surpasses more frequent loss of small reward. Indeed, the observed biases yielded increase of total reward (1.016 ± 0.001 times reward compared to no bias, mean ± SEM, slightly less than the optimal bias that yields 1.022 times reward compared to no bias), rather than maximizing the accuracy (=reward probability, i.e. no bias) or solely minimizing the risk (the variance of reward amounts) (*Figure 1D and E*).

Previous studies have shown that animals shift their decision boundary even without reward amount manipulations in perceptual decision tasks (*Lak et al., 2020a*). These shifts occur on a trial-by-trial basis, following a win-stay strategy, choosing the same side when that side was associated with reward in the previous trial, particularly when the stimulus was more ambiguous (*Lak et al., 2020a*). In the current task design, however, the optimal bias is primarily determined by the sizes of reward (more specifically, which side delivered a big or small reward) which stays constant across trials within a block. To determine whether the animal adopted short-time scale updating or a more stable bias, we next examined how receipt of reward affected the choice in the subsequent trials. To extract trial-by-trial updating, we compared the psychometric curves one trial before (n−1) and after (n+1) the current trials (n). This analysis was performed separately for the rewarded side in the current (n) trials. We found that choice biases before and after a specific reward location (choice in (n+1) trials minus choice in (n−1) trials) were not significantly different in any trial types (*Figure 1F*), suggesting that trial-by-trial updating was minimum, contrary to a previous study (*Lak et al., 2020b*). Notably, the previous study (*Lak et al., 2020b*) only examined the choice pattern in (n+1) trials to measure trial-by-trial updates, which was potentially overestimated because of global bias already seen in (n−1) trials. Instead, our results indicate that the mice adopted a relatively stable bias that lasts longer than one trial in our task.

Although we imposed a minimum time required to stay in the odor port, the mice showed different reaction times (the duration between odor onset and odor port exit) across different trial types (*Figure 1G*). First, reaction times were shorter when animals chose the BIG side compared to the

SMALL side in easy, but not difficult, trials. Second, reaction times were positively correlated with the level of sensory evidence for choice (as determined by odor % for the choice) when mice chose the BIG side. However, this modulation was not evident when mice chose the SMALL side.

Animals were required to stay in a water port for 1 s to obtain water reward. However, in rare cases, they exited a water port early, within 1 s after water port entry. We examined the effects of choice accuracy (correct or error) on the premature exit (*Figure 1H*). We found that while animals seldom exited a water port in correct trials, they occasionally exited prematurely in error trials, consistent with a previous study (*Kepecs et al., 2008*).

## Overall activity pattern of dopamine axons in the striatum

To monitor the activity of dopamine neurons in a projection-specific manner, we recorded the dopamine axon activity in the striatum using a calcium indicator, GCaMP7f (*Dana et al., 2019*) with fiber fluorometry (*Kudo et al., 1992*) (fiber photometry) (*Figure 2A*) ('dopamine axon activity' hereafter). We targeted a wide range of the striatum including the relatively dorsal part of VS, DMS and DLS (*Figure 2B*). Calcium signals were monitored from mice both before and after introducing water amount manipulations (n = 9, 7, six mice, for VS, DMS, DLS).

The main analysis was performed using the calcium signals obtained in the presence of water amount manipulations. To isolate responses that are time-locked to specific task events but with potentially overlapping temporal dynamics, we first fitted dopamine axon activity in each animal with a linear regression model using multiple temporal kernels (*Park et al., 2014*) with Lasso regularization with 10-fold cross validation (*Figure 2*). We used kernels that extract stereotypical time courses of dopamine axon activity locked to four different events: odor onset (odor), odor port exit (movement), water port entry (choice commitment or 'choice' for short), and reward delivery (water) (*Figure 2C–F*). Even if we imposed minimum 1 s delay, calcium signals associated with these events were potentially overlapped. In our task, the time course of events such as reaction time (odor onset to odor port exit) and movement time (odor port exit to water port entry) varied across trials. The model-fitting procedure finds kernels that best explain individual trial data in entire sessions assuming that calcium responses follow specific patterns upon each event and sum up linearly.

The constructed model captured modulations of dopamine axon activity time-locked to different events (*Figure 2C*). On average, the magnitude of the extracted odor-locked activity was modulated by odor cues. Dopamine axons were more excited by a pure odor associated with the BIG side than a pure odor associated with the SMALL side (*Figure 2C and F*). The movement-locked activity was stronger for a movement toward the contra-lateral (the opposite direction to the recorded hemisphere), compared to the ipsi-lateral side, which was most evident in DMS (*Parker et al., 2016*) but much smaller in VS or DLS (*Figure 2E*, % explained by movement). The choice-locked activity showed two types of modulations (*Figure 2C*). First, it exhibited an inhibition in error trials at the time of reward (i.e. when it has become clear that reward is not going to come). Second, dopamine axon activity showed a modulation around the time of water port entry, an excitation when the choice was correct, and an inhibition when the choice was incorrect, even before the mice received a feedback. These 'choice commitment'-related signals will be further analyzed below. Finally, delivery of water caused a strong excitation which was modulated by the reward size (*Figure 2C and F*). Furthermore, the responses to medium-sized water were slightly but significantly smaller on the BIG side compared to the SMALL side (*Figure 2C and F*). The contribution of water-locked kernels was larger than other kernels except in DMS, where odor, movement and water kernels contributed similarly (*Figure 2D and E*).

In previous studies, RPE-related signals have typically been characterized by phasic responses to reward-predictive cues and a delivery or omission of reward. Overall, the above results demonstrate that observed populations contain the basic response characteristics of RPEs. First, dopamine axons were excited by reward-predicting odor cues, and the magnitude of the response was stronger for odors that instructed the animal to go to the side associated with a higher value (i.e. BIG side). Responses to water were modulated by reward amounts, and the water responses were suppressed by higher reward expectation. These characteristics were also confirmed by using the actual responses, instead of the fitted kernels (*Figure 2F and G*). Finally, in error trials, dopamine axons were inhibited when the time passed beyond the expected time of reward, as the negative outcome becomes certain (*Figure 2C*). In the following sections, we will investigate each striatal area in more detail.

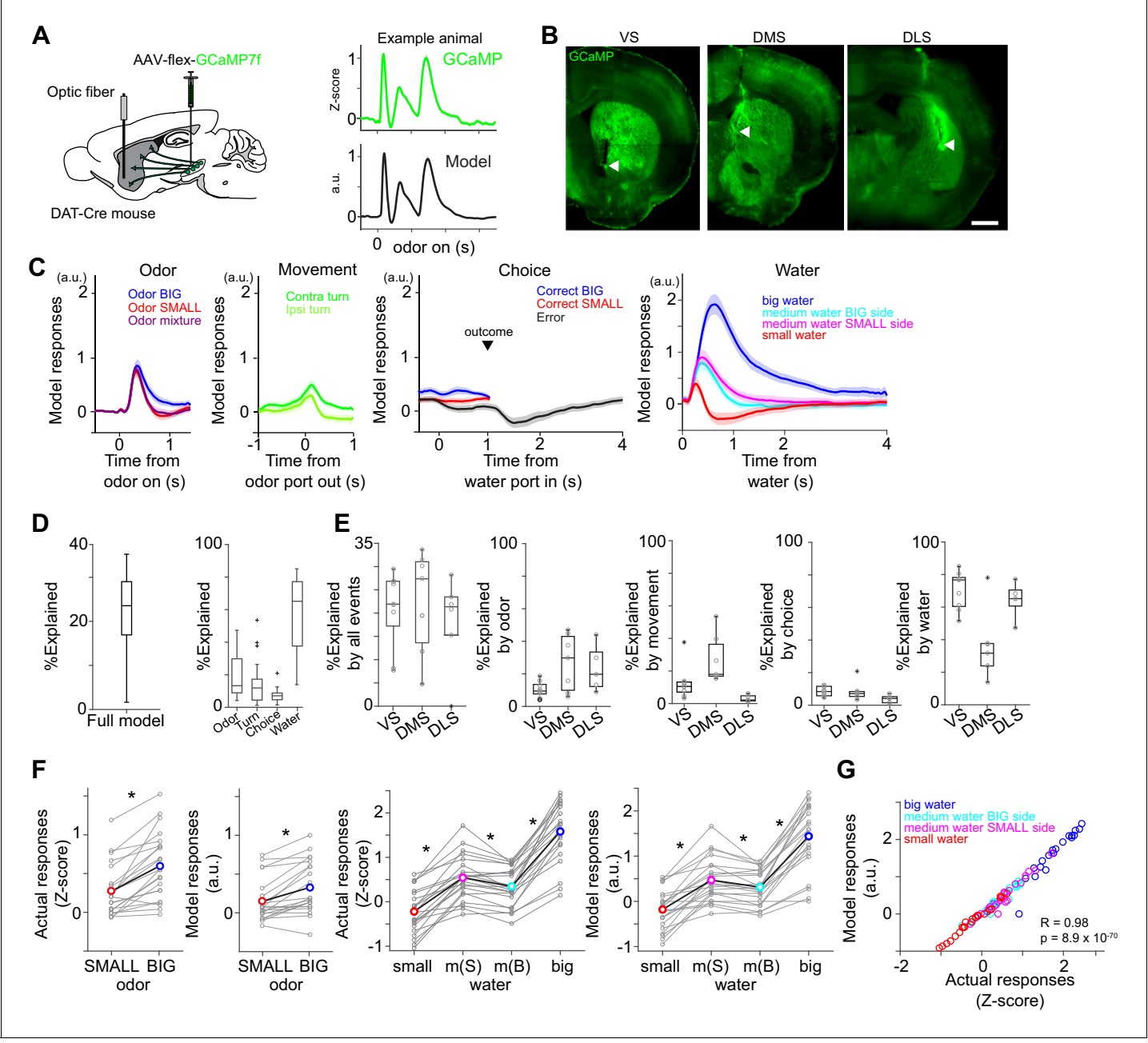

**Figure 2.** Dopamine axons in the striatum show characteristics of RPE. (A) AAV-flex-GCaMP7f was injected in VTA and SNc, and dopamine axon activity was measured with an optic fiber inserted in the striatum. Right top, dopamine axon activity in all the valid trials (an animal chose an either water port after staying in odor port for required time,>1 s) in an example animal, aligned at odor onset (mean ± SEM). Right bottom, average responses using predicted trial responses in a fitted model of the same animal (mean ± SEM). (B) Location of an optic fiber in example animals. Arrow heads, tips of fibers. Green, GCaMP7f. Bar = 1 mm. (C) Odor-, movement-, choice-, and water-locked components in the model of all the animals (mean ± SEM). (D) Contribution of each component in the model was measured by reduction of deviance in the full model compared to a reduced model excluding the component. (E) Contribution of each component in the model in each animal group. (F) Left, comparison of dopamine axon responses to an odor cue that instructs to choose BIG and SMALL side in easy trials (pure odor, correct choice, −1–0 s before odor port out). t(21) = 5.8, p=8.1 × 10⁻⁶ for actual signals and t(21) = 4.8, p=9.5 × 10⁻⁵ for models. Paired t-test, n = 22 animals. Right, comparison of dopamine axon responses to different sizes of water (big versus medium water with BIG expectation, and medium versus small water with SMALL expectation) and to medium water with different expectation (BIG versus SMALL expectation) (0.3–1.3 s after water onset). t(21) = 12.9, p=1.6 × 10⁻¹¹, t(21) = 9.7, p=2.9 × 10⁻⁹, and t(21) = −3.8, p=9.3 × 10⁻⁴, respectively for actual signals, and t(21) = 10.3, p=1.0 × 10⁻⁹, t(21) = 7.9, p=9.2 × 10⁻⁸, and t(21) = −3.3, p=0.0033, respectively for models. Paired t-test, n = 22 animals. m(B), medium water with BIG expectation; m(S), medium water with SMALL expectation. (G) Comparison between actual dopamine axon responses and model responses to water. Arbitrary unit (a.u.) was determined by model-fitting with z-score of GCaMP signals.

*Figure 2 continued on next page*

*Figure 2 continued*

The online version of this article includes the following source data and figure supplement(s) for figure 2:

**Source data 1.** Summary statistics.
**Figure supplement 1.** Dopamine axon activity outside of the task.

Dopamine activity is also modulated while a mouse moves without obvious reward (*Coddington and Dudman, 2018*; *Howe and Dombeck, 2016*; *da Silva et al., 2018*). To examine movement-related dopamine activity, we analyzed videos recorded during the task with DeepLab-Cut (*Mathis et al., 2018*; *Figure 2—figure supplement 1*). An artificial deep network was trained to detect six body parts: nose, both ears and three points along the tail – base, midpoint, and tip. To evaluate tracking in our task, we examined stability of a nose location detected by DeepLabCut when a mouse kept its nose in a water port, which was detected with the infra-red photodiode. After training of total 400 frames in 10 videos from 10 animals, error rates, calculated by disconnected tracking of nose position (50 pixel/frame), was $4.6 \times 10^{-4} \pm 1.5 \times 10^{-4}$ of frames (mean ± SEM, n = 43 videos), and nose tracking stayed within 2 cm when a mouse poked its nose into a water port for >1 s in 96.0% ± 0.3 of trials (mean ± SEM, n = 43 sessions) (*Figure 2—figure supplement 1B*). We examined dopamine axon activity when a mouse started or stopped locomotion (body speed is faster than at 3 cm/s), outside of the odor/water port area. In either case, we observed slightly but significantly lower dopamine axon activity level when a mouse moves, consistent with previous studies showing that some dopamine neurons show inhibition with movement (*Coddington and Dudman, 2018*; *Dodson et al., 2016*; *da Silva et al., 2018*; *Figure 2—figure supplement 1C,E*). We did not observe difference of modulation across the striatal areas (*Figure 2—figure supplement 1F*).

## Shifted representation of TD error in dopamine axon activity across the striatum

Although excitation to unpredicted reward is one of the signatures of dopamine RPE, recent studies found that the dopamine axon response to water is small or undetectable in some parts of the dorsal striatum (*Howe and Dombeck, 2016*; *Parker et al., 2016*; *da Silva et al., 2018*). Therefore, the above observation that all three areas (VS, DMS, and DLS) exhibited modulation by reward may appear at odds with previous studies.

We noticed greatly diminished water responses when the reward amount was not manipulated, that is, when dopamine axon signals were monitored during training sessions before introducing the reward amount manipulations (*Figure 3*, *Figure 3—figure supplement 1*). In these sessions, dopamine axons in some animals did not show significant excitation to water rewards (*Figure 3A and D*). This 'lack' of reward response was found in DMS, consistent with previous studies (*Parker et al., 2016*), but not in VS or DLS (*Figure 3G*). Surprisingly, however, DMS dopamine axons in the same animals showed clear excitation when reward amount manipulations were introduced, responding particularly strongly to a big reward (*Figure 3B and E*). Indeed, the response patterns were qualitatively similar across different striatal areas (*Figure 4*); the reward responses in all the areas were modulated by reward size and expectation, although the whole responses seem to be shifted higher in DLS, and lower in DMS (*Figure 4A and B*). These results indicate that the stochastic nature of reward delivery in our task enhanced or 'rescued' reward responses in dopamine axons in DMS.

The above results emphasized the overall similarity of reward responses across areas, but some important differences were also observed. Most notably, although a delivery of a small reward caused an inhibition of dopamine axons below baseline in VS and DMS, the activity remained non-negative in DLS. The overall responses tended to be higher in DLS.

In order to understand the diversity of dopamine responses to reward, we examined modulation of dopamine axon activity by different parameters (*Figure 4D*). First, the effect of the amount of 'actual' reward was quantified by comparing responses to different amounts of water for a given cue (i.e. the same expectation). The reward responses in all areas were modulated by reward amounts, with a slightly higher modulation by water amounts in VS (*Figure 4D* Water big-medium, Water medium-small). Next, the effect of expectation was quantified by comparing the responses to the same amounts of water with prediction of different amounts. Effects of reward size prediction were

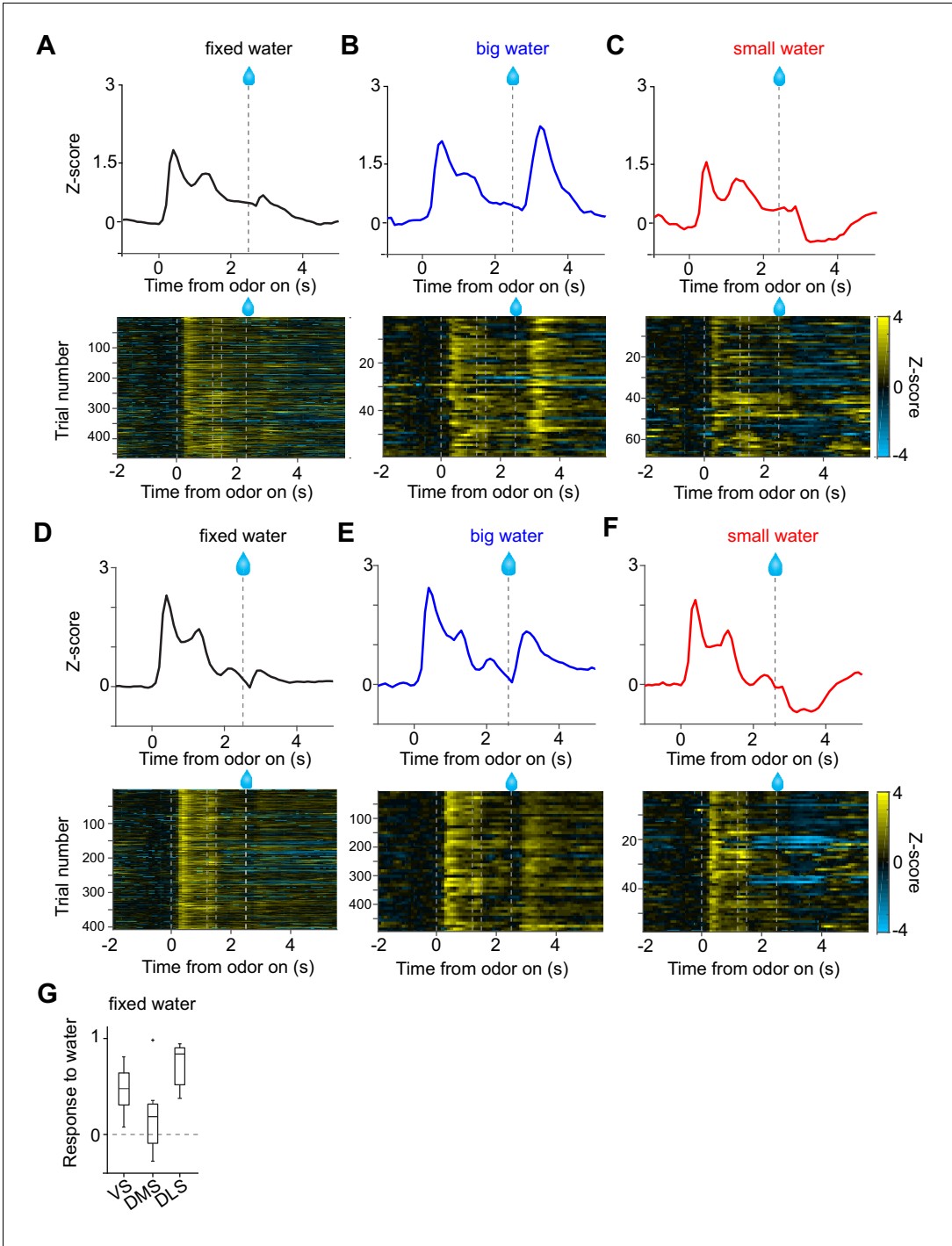

**Figure 3.** Small responses to fixed amounts of water in dopamine axons in DMS. (A, D) Dopamine axon responses to water in a fixed reward amount task (pure odor, correct choice). (B, E) Dopamine axon responses to a big amount of water in a variable reward amount task (pure odor, correct choice). (C, F) Dopamine axon responses to a small amount of water in a variable reward amount task (pure odor, correct choice). A-C, dopamine axon activity in an example animal; D-F, another example animal. (G) Responses to water (0.3–1.3 s after water onset) were significantly modulated with striatal location (F(2,19) = 5.1, p=0.016, ANOVA; t(11) = 2.9, p=0.013, DMS versus DLS; t(14) = 1.2, p=0.21, VS versus DMS; t(13) = −2.6, p=0.021, VS versus DLS, two sample t-test; t = 2.4, p=0.023, dorsal-ventral; t = −1.3, p=0.18, anterior-posterior; t = 1.6, p=0.10, medial-lateral, linear regression). The water responses were significantly positive in VS (t(8) = 4.7, p=0.0015) and in DLS (t(5) = 9.7, p=1.9 × 10$^{-4}$), but not in DMS (t(6) = 1.2, p=0.26). one sample t-test, n = 9, 7, six animals for VS, DMS, DLS.

*Figure 3 continued on next page*

*Figure 3 continued*

The online version of this article includes the following source data and figure supplement(s) for figure 3:

**Source data 1.** Summary statistics.

**Figure supplement 1.** Model responses in a task with fixed amounts of reward.

not significantly different across areas, although VS showed slightly less modulation with more variability (*Figure 4D*, prediction SMALL-BIG).

Next, we sought to characterize these differences between areas in simpler terms by fitting response curves (response functions). Previous studies that quantified responses of dopamine neurons to varied amounts of reward under different levels of expectation indicated that their reward responses can be approximated by a common function, with different levels of expectation just shifting the resulting curves up and down while preserving the shape (*Eshel et al., 2016*). We, therefore, fitted dopamine axon responses with a common response function (a power or linear function) for each expectation level (i.e. separately for BIG and SMALL) while fixing the shape of the function (i.e. the exponent of the power function or the slope of the linear function was fixed, respectively) (*Figure 4C*, *Figure 4—figure supplement 1A*). The obtained response functions for the three areas recapitulated the main difference between VS, DMS, and DLS, as discussed above. For one, the response curves of DLS are shifted overall upward. This can be characterized by estimating the amount of water that does not elicit a change in dopamine responses from baseline firing ('zero-crossing point' or reversal point). The zero-crossing points, obtained from the fitted curves, were significantly lower in DLS (*Figure 4C and D*). The results were similar regardless of whether the response function was a power (power function $\alpha<1$) or a linear function ($\alpha = 1$) (*Figure 4—figure supplement 1B*). Similar results were obtained using the aforementioned kernel models in place of the actual activity (*Figure 4—figure supplement 1D*).

Since the recording locations varied across animals, we next examined the relationship between recording locations and the zero-crossing points (*Figure 4E and F*). The zero-crossing points varied both along the medial-lateral and the dorsal-ventral axes (linear regression coefficient; $\beta = -50.8$ [zero-crossing point water amounts/mm], t = $-2.8$, p=0.011 for medial-lateral axis; $\beta = -43.1$, t = $-2.7$, p=0.014 for the dorsal-ventral axis). Examination of each animal confirmed that DMS showed higher zero-crossing points (upper-left in *Figure 4E* left) whereas DLS showed lower zero-crossing points (upper-right cluster in *Figure 4E* right).

We next examined whether the difference in zero-crossing points manifested specifically during reward responses or whether it might be explained by recording artifacts; upward and downward shifts in the response function can be caused by a difference in baseline activity before trial start (odor onset), and/or lingering activity of pre-reward activity owing to the relatively slow dynamics of the calcium signals (a combination of calcium concentration and the indicator). To examine these possibilities, the same analysis was performed after subtracting the pre-reward signals (*Figure 4—figure supplement 1C*). We observed similar or even bigger differences in zero-crossing points (F (2,19) = 20.5, p=1.7 $\times$ 10$^{-5}$, analysis of variance [ANOVA]). These results indicate that the elevated or decreased responses, characterized by different zero-crossing points, were not due to a difference in 'baseline' but were related to the difference that manifests specifically in responses to reward.

Considerably small zero-crossing points in dopamine axons in DLS were not due to a poor sensitivity to reward amounts nor a poor modulation by expected reward (*Figure 4D*). Different zero-crossing points, that is shifts of the boundary between excitation and inhibition at reward, suggest biased representation of TD error in dopamine axons across the striatum. In TD error models, difference in zero-crossing points may affect not only water responses but also responses to other events. Thus, the small zero-crossing points in dopamine axons in DLS should yield almost no inhibition following an event that is worse than predicted. To test this possibility, we examined responses to events with lower value than predicted (*Figure 5*): small water (*Figure 5A–C*), water omission caused by choice error (*Figure 5D–F*), and a cue that was associated with no outcome (*Figure 5G–I*). Consistent with our interpretation of small zero-crossing points, dopamine axons in DLS did not show inhibition in response to outcomes that were worse than predicted while being informative about water amounts.

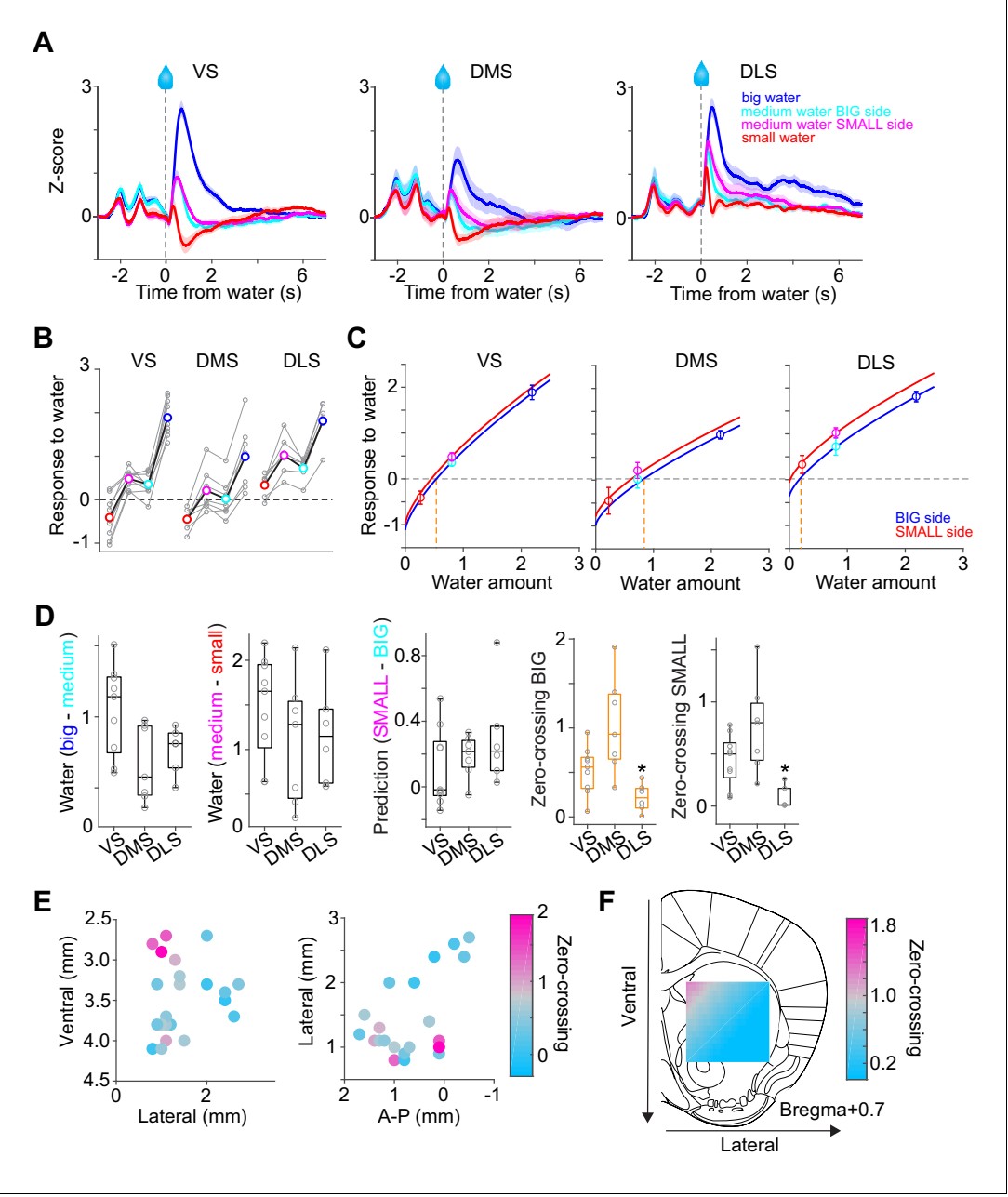

**Figure 4.** Responses to water in dopamine axons in the striatum. (**A**) Activity patterns per different striatal location, aligned at water onset (mean ± SEM, n = 9 for VS, n = 7 for DMS, n = 6 for DLS). (**B**) Average responses to each water condition in each animal grouped by striatal areas. (**C**) Average response functions of dopamine axons in each striatal area. (**D**) Comparison of parameters for each animal grouped by striatal areas. 'Water big-medium' is responses to big water minus responses to medium water at the BIG side and 'Water medium-small' is responses to medium water minus responses to small water at the SMALL side, normalized with difference of water amounts (2.2 minus 0.8 for BIG and 0.8 minus 0.2 for SMALL). 'Prediction SMALL-BIG' is responses to medium water at SMALL side minus responses to medium water at BIG side. 'Zero-crossing BIG' is the water amount when the dopamine response is zero at BIG and side, which was estimated by the obtained response function. 'Zero-crossing SMALL' is the water amount when the dopamine response is zero at SMALL side, which was estimated by the obtained response function. Response changes by water amounts (BIG or SMALL) or prediction was not significantly modulated by the striatal areas (F(2,19) = 4.33, p=0.028, F(2,19) = 0.87, p=0.43, F(2,19) = 1.11, p=0.34, ANOVA), whereas zero-crossing points (BIG or SMALL) were significantly modulated (F(2,19) = 8.6, p=0.0021, F (2,19) = 8.5, p=0.0023, ANOVA; t(11) = 3.6, p=0.0039, DMS versus DLS; t(14) = −2.4, p=0.028, VS versus DMS; t(13) = 2.4, p=0.030, VS versus DLS for BIG side; t(14) = −1.8, p=0.085, VS versus DMS; t(13) = 3.1, p=0.0076, VS versus

*Figure 4 continued on next page*

*Figure 4 continued*

DLS; t(11) = 3.88, p=0.0026, DMS versus DLS for SMALL side, two sample t-test). (**E**) Zero-crossing points were plotted along anatomical location in the striatum. Zero-crossing points were correlated with medial-lateral positions (t = −2.8, p=0.011) and with dorsal-ventral positions (t = −2.7, p=0.014) but not with anterior-posterior positions (t = −0.3, p=0.72). Linear regression. (**F**) Zero-crossing points were fitted with recorded location, and the estimated values in the striatal area were overlaid on the atlas for visualization (see Materials and methods). Trials with all odor types (pure and mixture) were used in this figure. t-test, n = 9, 7, six animals for VS, DMS, DLS. The online version of this article includes the following source data and figure supplement(s) for figure 4:

**Source data 1.** Summary statistics.
**Figure supplement 1.** Zero-crossing points across the striatum with different methods.

Taken together, these results demonstrate that dopamine reward responses in all three areas exhibited characteristics of RPEs. However, relative to canonical responses in VS, the responses were shifted more positively in the DLS and more negatively in the DMS.

## TD error dynamics in signaling perceptual uncertainty and cue-associated value

The analyses presented so far mainly focused on phasic dopamine responses time-locked to cues and reward. However, dopamine axon activity also exhibited richer dynamics between these events, which need to be explained. For instance, the signals diverged between correct and error trials even before the actual outcome was revealed (a reward delivery versus a lack thereof) (*Figure 2C* Choice). This difference between correct and error trials, which is dependent on the strength of sensory evidence (or stimulus discriminability), was used to study how neuronal responses are shaped by 'confidence'. Confidence is defined as the observer's posterior probability that their decision is correct given their subjective evidence and their choice ($P(reward|stimulus, choice)$) (*Hangya et al., 2016*). A previous study proposed a specific relationship between stimulus discriminability, choice and confidence (*Hangya et al., 2016*), although generality of the proposal is not supported (*Adler and Ma, 2018*; *Rausch and Zehetleitner, 2019*). Additionally, in our task, the mice combined the information about reward size with the strength of sensory evidence to select an action (confidence, or uncertainty) (*Figure 1*). The previous analyses did not address how these different types of information affect dopamine activity over time. We next sought to examine the time course of dopamine axon activity in greater detail, and to determine whether a simple model could explain these dynamics.

Our task design included two delay periods, imposed before choice movement and water delivery, to improve our ability to separate neuronal activity associated with different processes (*Figure 1A*). The presence of stationary moments before and after the actual choice movement allows us to separate time windows before and after the animal's commitment to a certain option. We examined how the activity of dopamine neurons changed before choice movement and after the choice commitment (*Figure 6*).

We first examined dopamine axon activity after water port entry (0–1 s after water port entry). In this period, the animals have committed to a choice and are waiting for the outcome to be revealed. Responses following different odor cues were plotted separately for trials in which the animal chose the BIG or SMALL side. The vevaiometric curve (a plot of responses against sensory evidence) followed the expected 'X-pattern' with a modulation by reward size (*Hirokawa et al., 2019*), which matches the expected value for these trial types, or the size of reward multiplied by the probability of receiving a reward, given the presented stimulus and choice (*Figure 6C*). The latter has been interpreted as the decision confidence, $P(reward|stimulus, choice)$ (*Lak et al., 2017*; *Lak et al., 2020b*). The crossing point of the two lines forming an 'X' is shifted to the left in our data because of the difference in the reward size (*Figure 6C*).

When this analysis was applied to the time period before choice movement (0–1 s before odor port exit), the pattern was not as clear; the activity was monotonically modulated by the strength of sensory evidence (%Odor BIG) only for the BIG choice trials, but not for the SMALL choice trials (*Figure 6B*). This result is contrary to a previous study that suggested that the dopamine activity reflecting confidence develops even before a choice is made (*Lak et al., 2017*). We note, however, that the previous study only examined the BIG choice trials, and the results were shown by 'folding' the x-axis, that is, by plotting the activity as a function of the stimulus contrast (which would

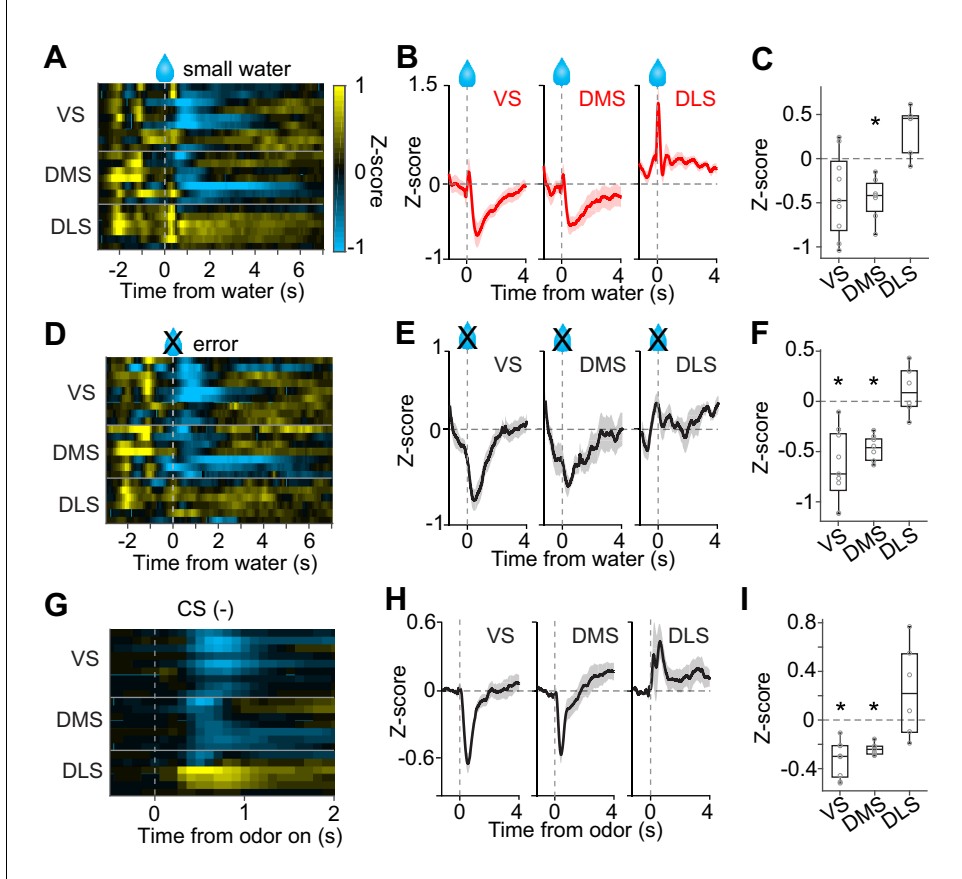

**Figure 5.** No inhibition by negative prediction error in dopamine axons in DLS. (**A**) Activity pattern in each recording site aligned at small water. (**B**) Average activity pattern in each brain area (mean ± SEM). (**C**) Mean responses to small water (0.3–1.3 s after water onset) were negative in VS and DMS (t(8) = −2.3, p=0.044; t(6) = −4.5, p=0.0040, responses versus baseline, one sample t-test), but not in DLS (t(5) = 3.3, p=0.020 responses versus baseline, one sample t-test). The responses were different across striatal areas (F(2,19) = 9.62, p=0.0013, ANOVA; t (13) = −3.4, p=0.0041, VS versus DLS; t(11) = −5.5, p=1.8 × 10$^{-4}$, DMS versus DLS: t(14) = 0.20, p=0.83, VS versus DMS, two sample t-test). (**D**) Activity pattern aligned at water timing in error trials. (**E**) Average activity pattern in each brain areas (mean ± SEM). (**F**) Mean responses in error trials (0.3–1.3 s after water timing) were negative in VS and DMS (t(8) = −5.4, p=6.2 × 10$^{-4}$; t(6) = −10.9, p=3.5 × 10$^{-5}$, responses versus baseline, one sample t-test), but not in DLS (t(5) = 1.1, p=0.30, responses versus baseline, one sample t-test). The responses were different across striatal areas (F(2,19) = 14.7, p=1.3 × 10$^{-4}$, ANOVA; t(13)=−4.5, p=5.6 × 10$^{-4}$, VS versus DLS; t(11)=-5.7, p=1.2 × 10$^{-4}$, DMS versus DLS; t(14) = −1.1, p=0.25, VS versus DMS, two sample t-test). (**G**) Activity pattern aligned at CS(-) in a fixed reward amount task. (**H**) Average activity pattern in each brain area (mean ± SEM). (**I**) Mean responses at CS(-) (−1–0 s before odor port out) were negative in VS and DMS (t(8) = −6.7, p=1.4 × 10$^{-4}$, VS; t(6) = −13.4, p=1.0 × 10$^{-5}$, DMS, responses versus baseline, one sample t-test), but not in DLS (t(5) = 1.5, p=0.17, responses versus baseline, one sample t-test). Responses were different across striatal areas (F(2,19) = 13.1, p=2.5 × 10$^{-4}$, ANOVA; t(13) = −4.1, p=0.0012, VS versus DLS; t(11) = −3.3, p=0.0065, DMS versus DLS; t(14) = −1.4, p=0.16, VS versus DMS, two sample t-test). n = 9, 7, six animals for VS, DMS, DLS.

The online version of this article includes the following source data for figure 5:

**Source data 1.** Summary statistics.

correspond to |%Odor BIG – 50| in our task), with the result matching the so-called 'folded X-pattern'. We would have gotten the same result, had we plotted our results in the same manner excluding the SMALL choice trials. Our results, however, indicate that a full representation of 'confidence' only becomes clear after a choice commitment, leaving open the question what the pre-choice dopamine axon activity really represents.

The aforementioned analyses, using either the kernel regression or actual activity showed that cue responses were modulated by whether the cue instructed a choice toward the BIG or SMALL

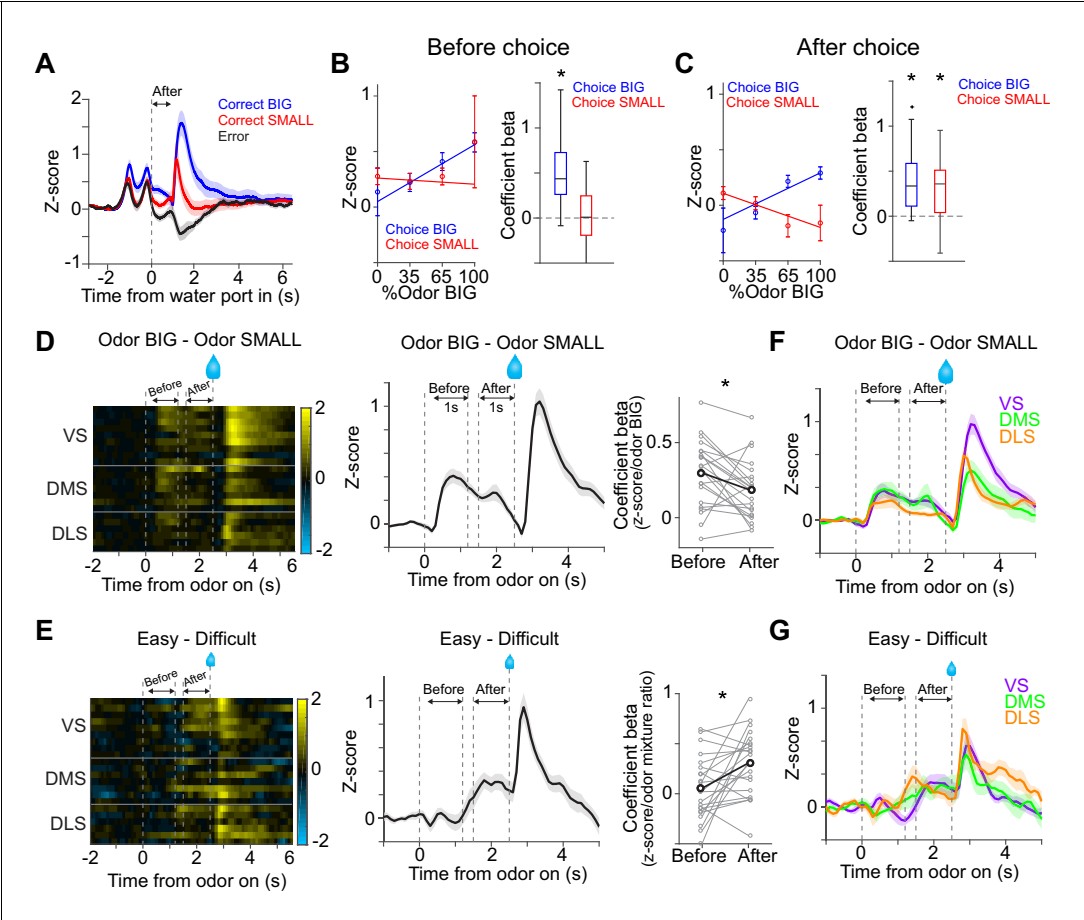

**Figure 6.** Dopamine signals stimulus-associated value and sensory evidence with different dynamics. (**A**) Dopamine axon activity pattern aligned to time of water port entry for all animals (mean ± SEM). (**B**) Responses before choice (−1–0 s before odor port out) were fitted with linear regression with odor mixture ratio, and coefficient beta (slope) for all the animals are plotted. Correlation slopes were significantly positive for choice of the BIG side (t (21) = 6.0, p=5.6 × 10$^{-6}$, one sample t-test), but not significant for choice of the SMALL side (t(21) = −0.8, p=0.42, one sample t-test). (**C**) Responses after choice (0–1 s after water port in) were fitted with linear regression with stimulus evidence (odor %) and coefficient beta (slope) for all the animals are plotted. Correlation slopes were significantly positive for both choice of the BIG side (t(21) = 5.6, p=1.4 × 10$^{-5}$, one sample t-test) and of the SMALL side (t(21) = 4.4, p=2.2 × 10$^{-4}$, one sample t-test). (**D**) Dopamine axon activity with an odor that instructed to choose BIG side (pure odor, correct choice) minus activity with odor that instructed to choose SMALL side (pure odor, correct choice) in each recording site (left), and the average difference in activity was plotted (mean ± SEM, middle). Correlation slopes between responses and stimulus-associated value (water amounts) significantly decreased after choice (t(21) = 2.3, p=0.026, before choice (−1–0 s before odor port out) versus after choice (0–1 s after water port in), pure odor, correct choice, paired t-test). (**E**) Dopamine axon activity when an animal chose SMALL side in easy trials (pure odor, correct choice) minus activity in difficult trials (mixture odor, wrong choice) in each recording site (left), and the average difference in activity was plotted (mean ± SEM, center). Coefficient beta between responses to odors and sensory evidence (odor %) significantly increased after choice (t(21) = −2.9, p=0.0078, before choice versus after choice, paired t-test). (**F**) Average difference in activity (odor BIG minus odor SMALL) before and after choice in each striatal area. The difference of coefficient (before versus after choice) was not significantly different across areas (F(2,19) = 0.15, p=0.86, ANOVA). (**G**) Average difference in activity (easy minus difficult) in each striatal area. The difference of coefficient (before versus after choice) was not significantly different across areas (F(2,19) = 1.46, p=0.25, ANOVA). n = 22 animals.

The online version of this article includes the following source data and figure supplement(s) for figure 6:

**Source data 1.** Summary statistics.

**Figure supplement 1.** Dopamine axon responses before and after choice in each striatal area.

side (*Figure 2C and F*). These results indicate that the information about stimulus-associated values (BIG versus SMALL) affected dopamine neurons earlier than the strength of sensory evidence (or confidence). We next examined the time course of how these two variables affected dopamine axon activity more closely. We computed the dopamine axon activity between trials when a pure odor instructed to go to the BIG versus SMALL side. Consistent with the above result, the difference was

evident during the cue period, and then gradually decreased after choice movement (*Figure 6D*). We performed a similar analysis, contrasting between easy and difficult trials (i.e. the strength of sensory evidence). We computed the difference between dopamine axon activity in trials when the animal chose the SMALL side after the strongest versus weaker stimulus evidence (a pure odor that instructs to choose the SMALL side versus an odor mixture that instructs to choose the BIG side). In stark contrast to the modulation by the stimulus-associated value (BIG versus SMALL), the modulation by the strength of stimulus evidence in SMALL trials fully developed only after a choice commitment (i.e. water port entry) (*Figure 6E*). Across striatal regions, the magnitude and the dynamics of modulation due to stimulus-associated values and the strength of sensory evidence were similar (*Figure 6F and G*), although we noticed that dopamine axons in DMS showed slightly higher correlation with sensory evidence before choice (*Figure 6—figure supplement 1*).

As discussed above, a neural correlate of 'confidence' appears at a specific time point (after choice commitment and before reward delivery) or in a specific trial type (when an animal would choose BIG side) before choice. We, therefore, next examined whether a simple model can account for dopamine axon activity more inclusively (*Figure 7*). To examine how the value and RPE may change within a trial, we employed a Monte-Carlo approach to simulate an animal's choices assuming that the animal has already learned the task. We used a Monte-Carlo method to obtain the ground truth landscape of the state values over different task states, without assuming a specific learning algorithm.

The variability and errors in choice in psychophysical performance are thought to originate in the variability in the process of estimating sensory inputs (perceptual noise) or in the process of selecting an action (decision noise). We first considered a simple case where the model contains only perceptual noise (*Green and Swets, 1966*). In this model, an internal estimate of the stimulus or a 'subjective odor' was obtained by adding Gaussian noise to the presented odor stimulus on a trial-by-trial basis (*Figure 7B* left). In each trial, the subject chooses deterministically the better option (*Figure 7C* left) based on the subjective odor and the reward amount associated with each choice (*Figure 7B* right). The model had different 'states' considering N subjective odors (N = 60 and 4 were used and yielded similar results), the available options (left versus right), and a sequence of task events (detection of odor, recognition of odor identity, choice movement, water port entry [choice commitment], Water/No-water feedback, inter-trial interval [ITI]) (*Figure 7A*). The number of available choices is two after detecting an odor but reduced to 1 (no choice) after water port entry. In each trial, the model receives one of the four odor mixtures, makes a choice, and obtains feedback (rewarded or not). After simulating trials, the state value for each state was obtained as the weighted sum of expected values of the next states, which was computed by multiplying expected values of the next states with probability of transitioning into the corresponding state. After learning, the state value in each state approximates the expected value of future reward, which is the sum of the amount of reward multiplied by probability of the reward (for simplicity, we assumed no temporal discounting of value within a trial). After obtaining state values for each state, state values for each odor ('objective' odor presented by experimenters) were calculated as the weighted sum of state values over subjective odors. After obtaining state values for each state for each objective odor, we then computed TD errors using a standard definition of TD error which is the difference between the state values at consecutive time points plus received rewards at each time step (*Sutton, 1987*).

We first simulated the dynamics of state values and TD errors when the model made a correct choice in easy trials, choosing either the BIG or SMALL side (*Figure 7F* bottom, blue versus red). As expected, the state values for different subjective odors diverged as soon as an odor identity was recognized, and the differences between values stayed constant as the model received no further additional information before acquisition of water. TD errors, which are the derivative of state values, exhibited a transient increase after odor presentation, and then returned to their baseline levels (near zero), remaining there until the model received a reward. Next, we examined how the strength of sensory evidence affected the dynamics of value and TD errors (*Figure 7F and J*). Notably, after choice commitment, TD error did not exhibit the additional modulation by the strength of sensory evidence, or a correlate of confidence (*Figure 7F* right and 7J right), contrary to our data (*Figure 7E and I* right). Thus, this simple model failed to explain aspects of dopamine axon signals that we observed in the data.

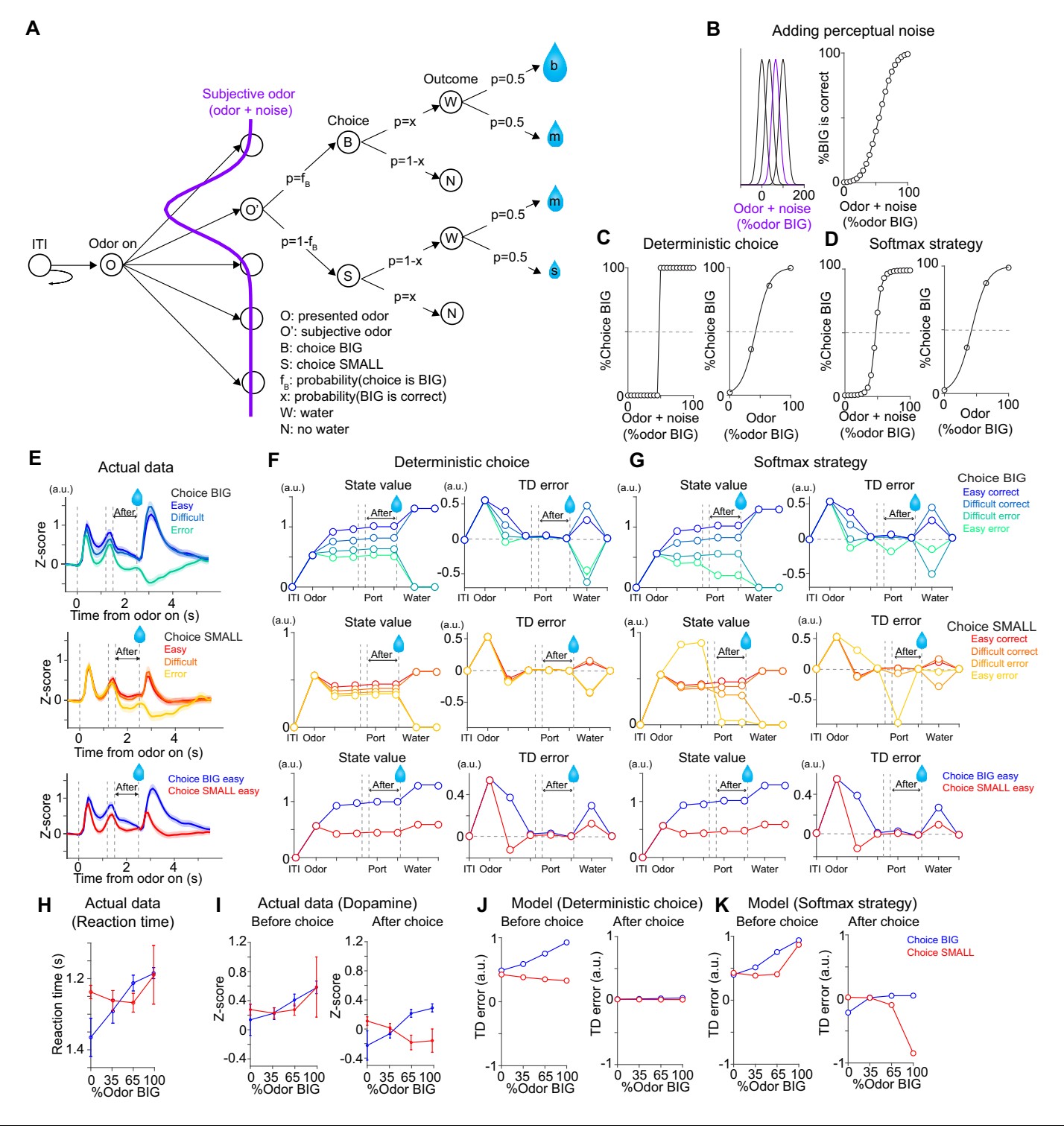

**Figure 7.** TD error dynamics capture emergence of sensory evidence after stimulus-associated value in dopamine axon activity. (**A**) Trial structure in the model. Some repeated states are omitted for clarification. (**B–D**) Models were constructed by adding perceptual noise with normal distribution to each experimenter's odor (B left, subjective odor), calculating correct choice for each subjective odor (B right), and determining choice for each subjective odor (C or D left) according to choice strategy in the model. The final choice for each objective odor by experimenters (odor %) was calculated as the weighted sum of choice for subjective odors (C or D right). (**E**) Dopamine axon activity in trials with different levels of stimulus evidence: easy (pure odor, correct choice), difficult (mixture odor, correct choice), and error (mixture odor, error), when animals chose the BIG side (top) and when animals chose the SMALL side (middle). Bottom, dopamine axon activity when animals chose the BIG or SMALL side in easy trials (pure odor, correct choice).

*Figure 7 continued on next page*

Figure 7 continued

(F, G) Time course in each trial of value (left) and TD error (right) of a model. (H) Line plots of actual reaction time from *Figure 1G*. Y-axis are flipped for better comparison with models. (I) Line plots of actual dopamine axon responses before and after choice from *Figure 6B and C*. (J, K) Model responses before and after choice were plotted with sensory evidence (odor %). Arbitrary unit (a.u.) was determined by value of standard reward as 1 (see Materials and methods).

The online version of this article includes the following figure supplement(s) for figure 7:

**Figure supplement 1.** TD errors with stochastic choice strategies.
**Figure supplement 2.** Comparison of model and actual responses.
**Figure supplement 3.** Correlation between dopamine axon signals and reaction time.
**Figure supplement 4.** Correlation between dopamine axon signals and movement time.
**Figure supplement 5.** Dopamine axon responses while animals stayed at water port.
**Figure supplement 6.** Dopamine axon signals and body movement when a mouse waits for water.

In the first model, we assumed that the model picks the best option given the available information in every trial (*Figure 7C*). In this deterministic model, all of the errors in choice are attributed to perceptual noise. We next considered a model that included decision noise in addition to the perceptual noise (*Figure 7D*). Here decision noise refers to some stochasticity in the action selection process, and may arise from errors in an action selection mechanism or exploration of different options, and can be modeled using different methods depending on rationale behind the noise. Here, we present results based on a 'softmax' decision rule, in which a decision variable (in this case, the difference in the ratio of the expected values at the two options) was transformed into the probability of choosing a given option using a sigmoidal function (e.g. Boltzmann distribution) (*Sutton and Barto, 1998*). We also tested other stochastic decision rules such as Herrnstein's matching law (*Herrnstein, 1961*) or $\varepsilon$-greedy exploration (randomly selecting an action in a certain fraction [$\varepsilon$] of trials) (*Sutton and Barto, 1998*; *Figure 7—figure supplement 1A–C*).

Interestingly, we were able to explain various peculiar features of dopamine axon signals described above simply by adding some stochasticity in action selection (*Figure 7G and K*). Note that the main free parameters of the above models are the width of the Gaussian noise, which determines the 'slope' of the psychometric curve, and was chosen based merely on the behavioral performance, but not the neural data. When the model chose the BIG side, state value at odor presentation was roughly monotonically modulated by the strength of sensory evidence similar to the above model (*Figure 7G* top left). When the model chose the SMALL side, however, the relationship between the strength of sensory evidence and value was more compromised (*Figure 7G* middle left). As a result, TD error did not show a monotonic relationship with sensory evidence before choice (*Figure 7G* middle right and 7K left), similar to actual dopamine axons responses (*Figure 7E* middle and 7I left), which was reminiscent of reaction time pattern (*Figure 7H*). On the other hand, once a choice was committed, the model exhibited interesting dynamics very different from the above deterministic model. After choice commitment, expected value was monotonically modulated by the strength of sensory evidence for both BIG and SMALL side choices (*Figure 7G* top and middle left, After). Further, because of the introduced stochasticity in action selection, the model sometimes chose a suboptimal option, resulting in a drop in the state value. This, in turn, caused TD error to exhibit an 'inhibitory dip' once the model 'lost' a better option (*Figure 7G* right), similar to the actual data (*Figure 7E and I*). This effect was strong particularly when the subjective odor instructed the BIG side but the model ended up choosing the SMALL side. For a similar reason, TD error showed a slight excitation when the model chose a better option (i.e. lost a worse option). The observed features in TD dynamics were not dependent on exact choice strategy: softmax, matching, and $\varepsilon$-greedy, all produced similar results (*Figure 7—figure supplement 1B,C*). This is because, with any strategy, after commitment of choice, the model loses another option with a different value, which results in a change in state value. These results are in stark contrast to the first model in which all the choice errors were attributed to perceptual noise (*Figure 7—figure supplement 2*, difference of Pearson's correlation with actual data p=0.0060, n = 500 bootstrap, see Materials and methods).

The observed activity pattern in each time window is potentially caused by physical movement. For example, the qualitative similarity of reaction time and dopamine axon activity before choice (*Figures 1G*, *6B* and *7H,I*) suggests some interaction. However, we did not observe trial-to-trial

correlation between reaction time and dopamine axon activity (*Figure 7—figure supplement 3*). On the other hand, we observed a weak correlation between movement time (from odor port exit to water port entry) and dopamine axon activity after choice (*Figure 7—figure supplement 4*). However, movement time did not show modulation by sensory evidence (*Figure 7—figure supplement 4B*) contrary to dopamine axon activity (*Figure 6C*), and dopamine axon activity was correlated with sensory evidence even after normalizing with movement time (*Figure 7—figure supplement 4C*). Since animals occasionally exit the water port prematurely in error trials (*Figure 1H*), we performed the same analyses as *Figure 6C* excluding trials where animals exited the water port prematurely. The results, however, did not change (*Figure 7—figure supplement 5*). While waiting for water after choice, the animal's body occasionally moved while the head stayed in a water port. We observed very small correlation between body displacement distances (body speed) and dopamine axon signals (*Figure 7—figure supplement 6*). However, this is potentially caused by motion artifacts in fluorometry recording, because we also observed significant correlation between dopamine axon signals and control fluorescence signals in each animal, although the direction was not consistent (*Figure 7—figure supplement 6A*). Importantly, neither body movement nor control signals showed modulation by choice accuracy (correct versus error) (*Figure 7—figure supplement 6B*). We performed linear regression of dopamine axon signals with body movement and accuracy with elastic net regularization, and dopamine axon signals were still correlated with accuracy (*Figure 7—figure supplement 6C*). These results indicate that the dopamine axon activity pattern we observed cannot be explained by gross body movement per se.

In summary, we found that a standard TD error, computing the moment-by-moment changes in state value (or, the expected future reward), can capture various aspects of dynamics in dopamine axon activity observed in the data, including the changes that occur before and after choice commitment, and the detailed pattern of cue-evoked responses. These results were obtained as long as we introduced some stochasticity in action selection (decision noise), regardless of how we did it. The state value dynamically changes during the performance of the task because the expected value changes according to an odor cue (i.e. strength of sensory evidence and stimulus-associated values) and the changes in potential choice options. A drop of the state value and TD error at the time of choice commitment occurs merely because the state value drops when the model chose an option that was more likely to be an error. Further, a correlate of 'confidence' appears after committing a choice, merely because at that point (and *only* at that point), the state value becomes equivalent to the reward size multiplied with the confidence, that is the probability of reward given the stimulus and the choice. This means that, as long as the animal has appropriate representations of states, a representation of 'confidence' can be acquired through a simple associative process or model-free reinforcement learning without assuming other cognitive abilities such as belief states or self-monitoring (meta-cognition). In total, not only the phasic responses but also some of the previously unexplained dynamic changes can be explained by TD errors computed over the state value, provided that the model contains some stochasticity in action selection in addition to perceptual noise. Similar dynamics across striatal areas (*Figure 6*) further support the idea that dopamine axon activity follows TD error of state values in spite of the aforementioned diversity in dopamine signals.

## Discussion

In the present study, we monitored dopamine axon activity in three regions of the striatum (VS, DMS and DLS) while mice performed instrumental behaviors involving perceptual and value-based decisions. In addition to phasic responses associated with reward-predictive cues and reward, we also analyzed more detailed temporal dynamics of the activity within a trial. We present three main conclusions. First, contrary to the current emphasis on diversity in dopamine signals (and therefore, to our surprise), we found that dopamine axon activity in all of the three areas exhibited similar dynamics. Overall, dopamine axon dynamics can be explained approximately by the TD error which calculates moment-by-moment 'changes' in the expected future reward (i.e. state value) in our choice paradigm. Second, although previous studies propose confidence as an additional variable in dopamine signals (*Engelhard et al., 2019*; *Lak et al., 2017*), correlates of confidence/choice accuracy naturally emerge in dynamics of TD error. Thus, mere observation of correlates of confidence in dopamine activity does not necessarily support that dopamine neurons multiplex information. Third, interestingly, however, our results showed consistent deviation from what TD model predicts. As

reported previously (*Parker et al., 2016*), during choice movements, contra-lateral orienting movements caused a transient activation in the DMS, whereas this response was negligible in VS and DLS. As pointed out in a previous study (*Lee et al., 2019*), this movement-related activity in DMS is unlikely to be a part of RPE signals. Nonetheless, dopamine axon signals overall exhibited temporal dynamics that are predicted by TD errors, yet, the representation of TD errors was biased depending on striatal areas. The activity during the reward period was biased toward positive responses in the DLS, compared to other areas; dopamine axon signals in DLS did not exhibit a clear inhibitory response ('dopamine dip') even when the actual reward was smaller than expected, or even when the animal did not receive a reward, despite our observations that dopamine axons in VS and DMS exhibited clear inhibitory responses in these conditions.

The positively or negatively biased reward responses in DLS and DMS can be regarded as important departures from the original TD errors, as it was originally formulated (*Sutton and Barto, 1998*). However, activation of dopamine neurons both in VTA and SNc are known to reinforce preceding behaviors (*Ilango et al., 2014*; *Keiflin et al., 2019*; *Lee et al., 2020*; *Saunders et al., 2018*), sharing, at least, their ability to function as reinforcement. Given the overall similarity between dopamine axon signals in the three areas of the striatum, these signals can be regarded as modified TD error signals. It is of note that our analyses are agnostic to how TD errors or underlying values are learned or computed: it may involve a model-free mechanism, as the original TD learning algorithm was formalized, or other mechanisms (*Akam and Walton, 2021*; *Langdon et al., 2018*; *Starkweather et al., 2017*). In any case, the different baselines in TD error-like signals that we observed in instrumental behaviors can provide specific constraints on the behaviors learned through dopamine-mediated reinforcement in these striatal regions.

## Confidence and TD errors

Recent studies reported that dopamine neurons are modulated by various variables (*Engelhard et al., 2019*; *Watabe-Uchida and Uchida, 2018*). One of such important variables is confidence or choice accuracy (*Engelhard et al., 2019*; *Lak et al., 2017*; *Lak et al., 2020b*). Distinct from 'certainty' that approximates probability broadly over sensory and cognitive variables, confidence often implies a metacognition process that specifically validates an animal's own decision (*Pouget et al., 2016*). Confidence can affect an animal's decision-making by modulating both decision strategy and learning. While there are different ways to compute confidence (*Fleming and Daw, 2017*), a previous study concluded that dopamine neurons integrate decision confidence and reward value information, based on the observation that dopamine responses were correlated with levels of sensory evidence (*Lak et al., 2017*). However, the interpretation is controversial since the results can be explained in multiple ways, for instance, with simpler measurements of 'difficulty' in a signal detection theory (*Adler and Ma, 2018*; *Fleming and Daw, 2017*; *Insabato et al., 2016*; *Kepecs et al., 2008*). More importantly, previous studies are limited in that (1) they focused on somewhat arbitrarily chosen trial types to demonstrate confidence-related activity in dopamine neurons (*Lak et al., 2017*), and that (2) they did not consider temporal dynamics of dopamine signals within a trial. Our analysis revealed that dopamine axon activity was correlated with sensory evidence only in a specific trial type and/or in a specific time window. At a glance, dopamine axon activity patterns may appear to be signaling distinct variables at different timings. However, we found that the apparently complex activity pattern across different trial types and time windows can be inclusively explained by a single quantity (TD error) in one framework (*Figure 7*). Importantly, the dynamical activity pattern became clear only if all the trial types were examined. We note that state value and sensory evidence roughly covary in a limited trial type (trials with BIG choice), while previous studies mainly focused on trials with BIG choice and responses in a later time window (*Lak et al., 2017*). Our results indicate that the mere existence of correlates of confidence or choice accuracy in dopamine activity was not evidence for coding of confidence, belief state or metacognition, as claimed in previous studies (*Lak et al., 2017*; *Lak et al., 2020b*) using a similar task as ours.

Whereas our model takes a primitive strategy to estimate state value, state value can be also estimated with different methods. The observed dopamine axon activity resembled TD errors in our model if agent's choice strategy is not deterministic (i.e. there is decision noise). However, confidence measurements in previous models *Hirokawa et al., 2019*, *Kepecs et al., 2008* and *Lak et al., 2017* used a fixed decision variable, and hence, did not consider dynamics and probability that animal's choice does not follow sensory evidence. A recent study proposed a different way of

computation of confidence by dynamically tracking states independent of decision variables (*Fleming and Daw, 2017*). A dynamical decision variable in a drift diffusion model also predicts occasional dissociation of confidence from choice (*van den Berg et al., 2016*; *Kiani and Shadlen, 2009*). While such dynamical measurements of confidence might be useful to estimate state value, confidence itself cannot be directly converted to state value because state value considers reward size and other choices as well. Interestingly, it was also proposed that a natural correlate of choice accuracy in primitive TD errors would be useful information in other brain areas to detect action errors (*Holroyd and Coles, 2002*). Together, our results and these models underscore the importance of considering moment-by-moment dynamics, and underlying computation.

## Similarity of dopamine axon signals across the striatum

Accumulating evidence indicates that dopamine neurons are diverse in many respects including anatomy, physiological properties, and activity (*Engelhard et al., 2019*; *Farassat et al., 2019*; *Howe and Dombeck, 2016*; *Kim et al., 2015*; *Lammel et al., 2008*; *Matsumoto and Hikosaka, 2009*; *Menegas et al., 2015*; *Menegas et al., 2017*; *Menegas et al., 2018*; *Parker et al., 2016*; *da Silva et al., 2018*; *Watabe-Uchida and Uchida, 2018*). Our study is one of the first to directly compare dopamine signals in three different regions of the striatum during an instrumental behavior involving perceptual and value-based decisions. We found that dopamine axon activity in the striatum is surprisingly similar, following TD error dynamics in our choice paradigm.

Our observation of similarity across striatal areas (*Figure 4A*) would give an impression that these results are different from previous reports. We note, however, that our ability to observe similarities in dopamine RPE signals depended on parametric variations of experimental parameters. For instance, if we only had sessions with equal reward amount on both sides (i.e. our training sessions), we might have concluded that DMS is unique in greatly lacking reward responses. However, this was not true: the use of probabilistic reward with varying amounts allowed us to reveal similar response functions across these areas as well as the specific difference (overall activation level). We also note that our results included movement-related activity which cannot readily be explained by TD errors (*Lee et al., 2019*), in particular, contra-lateral turn-related activity in DMS, consistent with a previous study (*Parker et al., 2016*), However, systematic examination of striatal regions showed that such movement-related activity was negligible in other areas such as DLS and VS. The turning movement is one of the most gross task-related movements in our task, yet, dopamine signals representing this movement were not wide-spread unlike TD error-like activity. Taken together, although we cannot exclude the possibility that dopamine activity in DLS is modulated by a specific movement in particular conditions, our results do not support that TD error-like activity in DLS is generated by a completely different mechanism or based on other types of information than other dopamine neuron populations.

Our results in DMS are consistent with previous studies that reported small and somewhat mysterious responses to reward (*Brown et al., 2011*; *Parker et al., 2016*). We noticed that while animals were trained with fixed amounts of water, some of the dopamine axon signals in DMS did not exhibit clear responses to water, and on average water responses were smaller than in other areas (*Figure 3*, *Figure 3—figure supplement 1*). Once reward amounts became probabilistic, dopamine axons in DMS showed clear responses according to RPE (*Figure 3*, *Figure 4*), similar to the previous observation that dopamine responses to reward in DMS emerged after contingency change (*Brown et al., 2011*). Why are reward responses in DMS sometimes observed and sometimes not? We found that the response function for water delivery in dopamine axons in different striatal areas showed different zero-crossing points, the boundary between excitatory and inhibitory responses (*Figure 4*). The results suggested that dopamine axons in DMS use a higher standard (requiring larger amounts of reward to excite). In other words, dopamine signals in DMS use a strict criterion to be excited. Higher criteria in DMS may partly explain the observation that some dopamine neurons do not show a clear excitation by reward, such as in the case of our recording without reward amount modulations (*Figure 3A,D,G*). However, considering that dopamine responses to free water were also negligible in DMS in some studies (*Brown et al., 2011*; *Howe and Dombeck, 2016*), whether dopamine neurons respond to reward likely depends critically on task structures and training history. One potential idea is that dopamine in DMS has a higher excitation threshold because the system predicts upcoming reward optimistically, along not only size but also time, causing smaller RPE (predicting away) easily with little evidence. Optimistic expectation echoes with the idea

of Watkin's Q-learning algorithm (*Watkins, 1989*; *Watkins and Dayan, 1992*) where an agent uses the maximum value among values of potential actions to compute RPEs, although we did not explore action values explicitly in this study. Future studies are needed to find the functional meaning of optimism in dopamine neurons and to examine whether the optimism is responsible for a specific learning strategy in DMS. We also have to point out that because fluorometry in our study only recorded average activity of dopamine axons, we likely missed diversity within dopamine axons in a given area. It will be important to further examine in what conditions these dopamine neurons lose responses to water, or whether there are dopamine neurons which do not respond to reward in any circumstances.

In contrast to DMS, we observed reliable excitation to water reward in dopamine axons in DLS. However, because we only recorded population activity of dopamine axons, our results do not exclude the possibility that some dopamine neurons that do not respond to reward also project to DLS. Alternatively, the previous observation that some dopamine neurons in the substantia nigra show small or no excitation to reward (*da Silva et al., 2018*) may mainly come from DMS-projecting dopamine neurons or another subpopulation of dopamine neurons that project to the tail of the striatum (TS) (*Menegas et al., 2018*), but not DLS. Notably, the study (*da Silva et al., 2018*) also used predictable reward (fixed amounts of water with 100% contingency) to examine dopamine responses to reward. In contrast, we found that dopamine axons in DLS show strong modulation by reward amounts and prediction, and their dynamics resemble TD errors in our task. Our observation suggests that the lack of reward omission responses and excitation by even small rewards in instrumental tasks is key for the function of dopamine in DLS.

## Positively biased reinforcement signals in DLS dopamine

It has long been observed that the activity of many dopamine neurons exhibits a phasic inhibition when an expected reward is omitted or when the reward received is smaller than expected (*Hart et al., 2014*; *Schultz et al., 1997*). This inhibitory response to negative RPEs is one of the hallmarks of dopamine RPE signals. Our results that dopamine axon signals in DLS largely lack these inhibitory dips (*Figure 4* and *Figure 5*) has profound implications on what types of behaviors are learned through DLS dopamine signals as well as what computational principles underlie reinforcement learning in DLS.

Dopamine 'dips' are thought to act as aversive stimuli and/or can facilitate extinction of previously learned behaviors (weakening) (*Chang et al., 2018*; *Montague et al., 1996*; *Schultz et al., 1997*). The lack of dopamine dip in DLS may lead to the animal's reduced sensitivity to worse-than-expected outcome (i.e. negative prediction error). This characteristic resembles the activity of dopamine axons in TS, posterior to DLS, which signals potential threat and also lacks inhibitory responses to an omission of a predicted threat (*Menegas et al., 2017*; *Menegas et al., 2018*). We proposed that the lack of inhibitory omission signals (and so lack of weakening signals) would be critical to maintain threat prediction even if an actual threat is sometimes omitted. Similarly, the lack of weakening signals in DLS may help keep the learned actions from being erased even if the outcome is sometimes worse than predicted or even omitted. This idea is in line with the previous observations that DLS plays an important role in habitual behaviors (*Yin et al., 2004*). The uniquely modified TD error signal in DLS (i.e. a reduced inhibitory response during the reward period) may explain a predominant role of DLS in controlling habitual behaviors.

(*Thorndike, 1932*) proposed two principles for instrumental learning – the law of effect and the law of exercise. The law of effect emphasizes the role of outcome of behaviors: behaviors that led to good outcomes become more likely to occur – an idea that forms the basis of value-based reinforcement learning. In contrast, the law of exercise emphasizes the number of times a particular action was taken. There has been an increasing appreciation of the law of exercise because repetition or overtraining is the hallmark of habits and skills (*Hikosaka et al., 1995*; *Matsuzaka et al., 2007*; *Miller et al., 2019*; *Morris and Cushman, 2019*; *Ölveczky, 2011*; *Robbins and Costa, 2017*; *Smith and Graybiel, 2016*), whereas reinforcement learning models address the law of effect.

The clear deviation from TD error in dopamine signals in DLS, the lack of inhibitory dip with negative prediction error, revises existing reinforcement learning models in the basal ganglia that assume the same teaching signals across the striatum. On the other hand, recent studies pointed out that the basic reinforcement learning models do not explain the function of DLS, proposing different mechanisms such as value-less teaching signals to support the law of exercise (*Dezfouli and*

Balleine, 2012; Miller et al., 2019). Here, we propose that dopamine signals in DLS provide an ideal neural substrate of learning with an emphasis on the law of exercise. A positively biased TD error signal ensures that an 'OK' action will be positively reinforced, in a manner that depends on the number of times that the same behavior was repeated as far as it is accompanied by a small reward (i.e. with 'OK' signals). This property may explain why the formation of habit (and skills) normally requires overtraining (i.e. repeating a certain behavior many times).

The observation that DLS dopamine signals lack inhibitory responses raises the question what is actually learned by the system. Learning of values depends on the balance between positive and negative prediction errors: the learned value converges to the point at which positive and negative prediction errors form an equilibrium. If a reinforcement signal lacks negative prediction errors, this learning would no longer work as it was originally conceptualized. In reinforcement learning theories, an alternative approach is policy-based reinforcement learning, learning of 'preference' (Sutton and Barto, 2018) rather than value. One way to conceptualize preference is to see it as a generalized version of value, which has less constraints than value (the idea of 'value' may imply many properties that it should follow, for example the value should be zero for no outcome). We propose that policy learning may be a better way to conceptualize the function of the DLS, as was proposed in previous studies (Sutton and Barto, 1998; Takahashi et al., 2008). Instead of finding an optimal solution seen in policy gradient methods (Sutton and Barto, 2018), positively-biased TD errors in DLS may directly reinforce taken actions as proposed originally (Barto et al., 1983; Sutton and Barto, 1998), thus preserving the law of effect but also emphasize the law of exercise. Considering that the main inputs to DLS come from the motor cortex, somatosensory cortex, and other subcortical areas such as intralaminar nuclei in thalamus (Hunnicutt et al., 2016), positively biased teaching signals potentially play a role in chaining actions by training.

In summary, we propose that the learning of habits and skills are a natural consequence of reinforcement learning using a specialized reinforcement signal (positively shifted response to outcomes) and the unique anatomical property (the specialized input of motor and somatosensory information) of the DLS. Future experiments using tasks involving sequence of actions (Hikosaka et al., 1995; Ölveczky, 2011) can test this idea.

## Potential mechanisms underlying diverse TD error signals

We found that, across the striatum, dopamine signals overall resemble TD errors, with positive or negative bias in a subregion-specific manner (Figure 4). How such a diversity is generated is an open question. One potential mechanism is by optimistic and pessimistic expectations, as proposed in distributional reinforcement learning (Dabney et al., 2020; Lowet et al., 2020). A recent study (Dabney et al., 2020) proposed that the diversity in dopamine responses potentially give rise to a population code for a reward distribution (distributional reinforcement learning). In this theory, there are optimistic and pessimistic dopamine neurons. Optimistic dopamine neurons emphasize positive over negative RPEs, and as a consequence, their corresponding value predictors are biased to predict a higher value in a reward distribution, or vice versa. The distributional reinforcement learning, as formulated in Dabney et al., 2020, predicts that optimistic and pessimistic dopamine neurons should have zero-crossing points shifted toward larger and smaller rewards, respectively. In this sense, our observation that DLS dopamine signals have smaller zero-crossing points resembles pessimistic dopamine neurons in distributional reinforcement learning, although the previous study found both optimistic and pessimistic dopamine neurons in the VTA, which does not necessarily project to the DLS. Whether the present result is related to distributional reinforcement learning requires more specific tests such as dopamine neurons' sensitivity to positive versus negative RPEs (Dabney et al., 2020). It will be interesting to characterize these response properties in a projection-specific manner.

Alternatively, DLS-projecting dopamine neurons may add 'success premium' at each feedback. Signals of success feedback were observed in multiple cortical areas (Chen et al., 2017; Sajad et al., 2019; Stuphorn et al., 2000), which is often more sustained than phasic dopamine responses. Interestingly, we noticed that responses to water in dopamine axons in DLS are more sustained than dopamine axons in other areas (Figure 4A). DLS-projecting dopamine neurons potentially receive and integrate those success feedback signals with reward value, shifting the teaching signals more positively.

Mechanistically, biases in dopamine signals may stem from a difference in the excitation-inhibition balance at the circuit level. In addition to dopamine neurons, there are multiple brain areas where activity of some neurons resembles RPE (*Li et al., 2019*; *Matsumoto and Hikosaka, 2007*; *Oyama et al., 2010*; *Tian et al., 2016*). Among these, presynaptic neurons in multiple brain areas directly convey a partial prediction error to dopamine neurons (*Tian et al., 2016*). On the other hand, the rostromedial tegmental area (RMTg) exhibits a flipped version of RPE (the sign is opposite to dopamine neurons), and its inhibitory neurons directly project to dopamine neurons in a topographic manner (*Hong et al., 2011*; *Jhou et al., 2009a*; *Jhou et al., 2009b*; *Li et al., 2019*; *Tian et al., 2016*). Hence, each dopamine neuron may receive a different ratio of excitatory and inhibitory inputs of RPE. Interestingly, previous studies found that inactivation of neurons in RMTg (or habenula, its input source) mainly affected dopamine responses to negative events even though these neurons represent both positive and negative RPE (*Li et al., 2019*; *Tian and Uchida, 2015*). Based on these findings, we propose a model in which different ratios of TD error signals in presynaptic neurons cause different zero-crossing points in dopamine subpopulations. We simulated how different ratios of TD error inputs may affect output TD error signals (*Figure 8*). We simplified a model with only two inputs, excitatory and inhibitory, that have stronger effects on postsynaptic neurons with excitation than inhibition (*Figure 8A*). We found that just having different ratios of inputs can cause different zero-crossing points (*Figure 8B,C*) because of the dynamic pattern of dopamine activity (detection and discrimination) in an overlapped time window (*Nomoto et al., 2010*). This mechanistic model is consistent with previous findings that distribution of presynaptic neurons to projection-specific dopamine subpopulations are similar to each other but quantitatively slightly different (*Beier et al., 2015*; *Lerner et al., 2015*; *Menegas et al., 2015*). It would be interesting if DLS-projecting dopamine neurons receive less inhibitory RPE, and DMS-projecting dopamine neurons receive more, so that RPE signals are pushed up or down, whereas the information is still almost intact (*Figure 8D*). It is important to examine whether these dopamine neurons show detectable inhibition with large negative prediction error such as actual reward omission in an easy task, as the model predicts. In addition to anatomical reasons, DLS-projecting dopamine neurons show higher burstiness in intact animals (*Farassat et al., 2019*) and higher excitability in vitro (*Evans et al., 2017*; *Lerner et al., 2015*). These multiple reasons may explain why DLS-projecting dopamine neurons do not show inhibitory responses to negative prediction errors. It will be fascinating if we can connect all these levels of studies into functional meaning in the future.

## Limitations and future directions

This study is one of the first systematic comparisons of dopamine axon activity across the striatum using parametric decision-making task. Although we tried to target various locations along anterior-posterior, dorsal-ventral, and medial-lateral axes (*Figure 4E,F*), we did not cover the entire striatum such as the most posterior parts (TS) and ventral portions of VS including the medial shell of the nucleus accumbens. Multiple studies reported unique natures of dopamine activity in these areas (*Brown et al., 2011*; *Lammel et al., 2008*; *Menegas et al., 2018*). It is important to include these areas and to examine whether the observed difference in zero-crossing points is gradual or defined by a boundary, and to determine the boundary if there is.

Our task incorporated a typical perceptual task using different levels of sensory evidence (*Rorie et al., 2010*; *Uchida and Mainen, 2003*) into a value-based learning with probabilistic reward manipulation using 2 sets of different sizes of water. Although the task was demanding to mice, we were able to detect RPE natures in dopamine axon activity without over-training. However, the difference of prediction BIG versus SMALL sides was still small. Further, most analyses relied on pooled data across sessions because of the limited number of trials in each trial type, especially error trials. Further improvement of the task will facilitate more quantitative analyses over learning.

In this study, we modeled dynamical representation patterns of dopamine neurons in a steady state, but did not examine relationship between dopamine activity and actual learning. Especially, while our model used a discrete single stimulus state in each trial, it is naturalistic that animals use the experience from a single trial to update value in other stimulus states (*Bromberg-Martin et al., 2010*), and/or animals represent states in a more continuous manner (*Kiani and Shadlen, 2009*). It is important in the future to examine how dynamical and diverse dopamine signals are used during learning and/or performance.

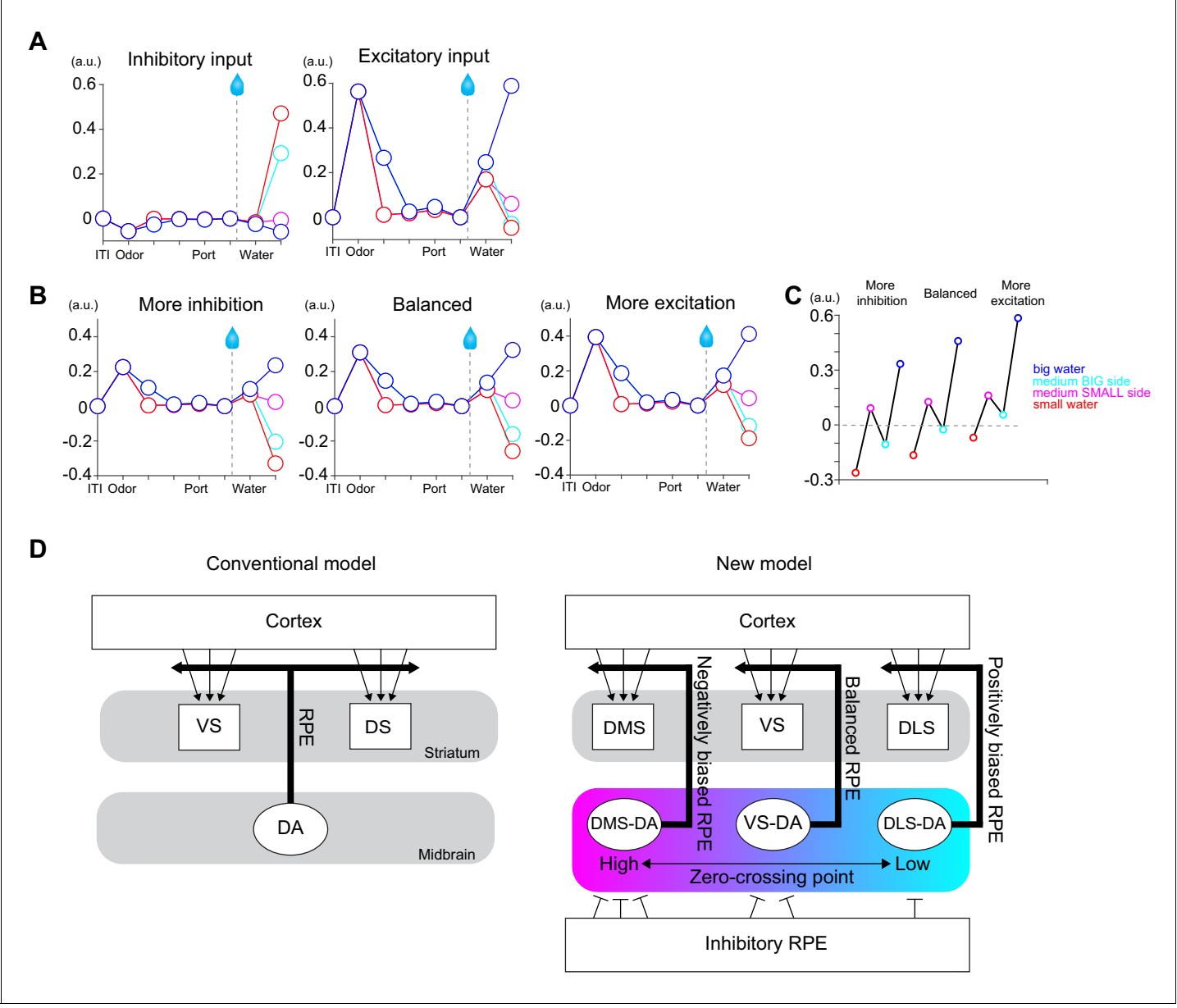

**Figure 8.** A potential mechanism of different zero-crossing points in dopamine neurons. (**A**) A simplified model with only two inputs, one is inhibitory, and the other is excitatory, both of which encode TD errors but send information to the postsynaptic neurons mainly with excitation (1/10 with inhibition). (**B**) TD errors in postsynaptic neurons with different ratios of presynaptic inputs, balanced (1:1), more inhibition (two times more inhibitory inputs) and more excitatory (half of inhibitory inputs). (**C**) Net responses to water in these three postsynaptic neurons. (**D**) Left, conventional models such as actor-critic models assume the same TD errors to be broadcasted throughout the striatum. Right, we propose that the striatal subareas receive slightly different TD errors with different zero-crossing points. One of potential mechanisms is different ratios of presynaptic inputs. Arbitrary unit (a.u.) was determined by value of standard reward as 1 (see Materials and methods).

Multiple studies suggested a close relationship between dopamine signaling and movement (*Coddington and Dudman, 2018*; *Howe and Dombeck, 2016*; *da Silva et al., 2018*). While we observed a slight inhibition of dopamine axon signals with locomotion outside of the task across the striatal subareas (*Figure 2—figure supplement 1*), we must exercise caution while interpreting this result. First, our task is not designed to address effects of movement. Even when animals were outside of the port area, animals potentially engaged in rewarding actions such as drinking water remaining in the mouth, eating feces and grooming. Further, we observed weak but significant motion artifacts in control fluorescence signals in fluorometry signals (*Figure 7—figure supplement*

6). Further studies using more precise behavioral observation (*Mathis and Mathis, 2020*; *Wiltschko et al., 2020*) and motion-resistant recording techniques are needed to understand movement-related dopamine activity in the striatal subareas in freely moving animals.

Taken together, our results showed that dopamine axon signals in the striatum approximate TD error dynamics. We propose that dopamine in different striatal areas conveys TD errors in a biased manner. One compelling idea is that the lack of negative teaching signals in DLS plays a role in skill/habit, although further examination is needed to establish its functions. We also observed some deviation from TD errors such as contra-lateral turn-related activity in DMS and slight inhibition with locomotion. It is important to test these other parameters in the future in order to understand the meaning of the diversity of dopamine neurons and organization of dopamine-striatum systems.

# Materials and methods

Key resources table

| Reagent type (species) or resource | Designation | Source or reference | Identifiers | Additional information |
|---|---|---|---|---|
| Transgenic mouse strain | Dopamine transporter (DAT)-cre | Jackson laboratory | B6.SJL-Slc6a3tm1.1 (cre)Bkmn/J | RRID:IMSR JAX:006660 |
| Transgenic mouse strain | Ai14 | Jackson laboratory | Rosa-CAG-LSL-tdTomato | RRID:IMSR JAX:007914 |
| Virus strain | GCaMP7f | UNC Vector Core | AAV5-CAG-FLEX-GCaMP7f | $1.8 \times 10^{13}$ particles/ml |
| Virus strain | tdTomato | UNC Vector Core | AAV5-CAG-FLEX-tdTomato | $2.0 \times 10^{13}$ particles/ml |

## Animals

17 dopamine transporter (DAT)-cre (B6.SJL-Slc6a3tm1.1(cre)Bkmn/J, Jackson Laboratory; RRID:IMSR JAX:006660) (*Bäckman et al., 2006*) heterozygous mice, and 5 DAT-Cre;Ai14 (Rosa-CAG-LSL-tdTomato, Jackson Laboratory; RRID:IMSR JAX:007914) (*Madisen et al., 2010*) double heterozygous mice, male and female, were used for recording signals from dopamine axons. All mice were backcrossed with C57BL/6J (Jackson Laboratory). Animals were housed on a 12 hr dark/12 hr light cycle (dark from 07:00 to 19:00) and performed a task at the same time each day. Animals were group-housed (2–4 animals/cage) during training, and then single-housed after surgery. All procedures were performed in accordance with the National Institutes of Health Guide for the Care and Use of Laboratory Animals and approved by the Harvard Animal Care and Use Committee.

## Surgical procedures

All surgeries were performed under aseptic conditions with animals anesthetized with isoflurane (1–2% at 0.5–1.0 l/min). Analgesia was administered pre (buprenorphine, 0.1 mg/kg, I.P) and postoperatively (ketoprofen, 5 mg/kg, I.P). To express GCaMP7f (*Dana et al., 2019*) specifically in dopamine neurons, we unilaterally injected 300 nl of mixed virus solution; AAV5-CAG-FLEX-GCaMP7f ($1.8 \times 10^{13}$ particles/ml, UNC Vector Core, NC) and AAV5-CAG-FLEX-tdTomato ($2.0 \times 10^{13}$ particles/ml, UNC Vector Core, NC) into both the VTA and SNc (600 nl total) in the DAT-cre mice. Only AAV5-CAG-FLEX-GCaMP7f (300 nl total) was used for DAT;Ai14 double transgenic mice. Virus injection lasted around 20 min, and then the injection pipette was slowly removed over the course of several minutes to prevent damage to the tissue. We also implanted optic fibers (400 μm diameter, Doric Lenses, Canada) into the VS, DMS, or DLS (one fiber per mouse). To do this, we first slowly lowered optical fibers into the striatum. Once fibers were lowered, we first attached them to the skull with UV-curing epoxy (NOA81, Thorlabs, NJ), and then a layer of rapid-curing epoxy to attach the fiber cannulas even more firmly to the underlying glue. After waiting 15 min for this to dry, we applied a black dental adhesive (Ortho-Jet, Lang Dental, IL). We used magnetic fiber cannulas (Doric Lenses, MFC_400/430) and the corresponding patch cords to allow for recordings in freely moving animals. After waiting 15 min for the dental adhesive to dry, the surgery was complete. We used the following coordinates to target our injections and implants.

- (VTA) Bregma: −3.0 mm, Lateral: 0.6 mm, Depth: between 4.5 mm and 4.3 mm
- (SNc) Bregma: −3.0 mm, Lateral: 1.6 mm, Depth: between 4.3 mm and 4.1 mm
- (VS) Bregma: between 1.5 and 1.0 mm, Lateral: 1.8 mm, Depth: 3.8 mm, angle 10°

- (DMS) Bregma: between 1.5 and 0 mm, Lateral: 1.3 mm, Depth: 2.3 mm
- (DLS) Bregma: between 1.3 and −0.8 mm, Lateral: 3.0 mm, Depth: 2.3 mm

## Behavioral tasks

The behavioral apparatus consisted of a custom-built behavioral box (32 × 19 × 30 cm) (*Figure 2—figure supplement 1A*) containing three conical nose-pokes (38 mm inner diameter, 38 mm depth). The odor port was located in the middle of one wall (19 cm) at a height of 27 mm from the floor to center. Two choice ports were located 45 mm left and right of the odor port at a 45° angle. An infra-red photodiode/phototransistor pair placed on either side of the nose poke at 15 mm depth from the surface was used to determine the timing of nose pokes.

All behavioral experiments were controlled by a NIDAQ board (National Instruments, TX) and Labview (National Instruments, TX), similar to a previous study (*Uchida and Mainen, 2003*). Mice were trained to perform an odor-discrimination task for water reward, similar to a study in rats (*Uchida and Mainen, 2003*) with several modifications. Mice initiated trials in a self-paced manner by poking a center port, which then delivered an odor. Different odors were used in a pseudor-andomized order from three different pure chemicals (odor A, B, and C) and mixtures of odor A and B with various ratios. Mice were required to choose a left or right water port depending on dominant odor identity, odor A or B. Correct choice was always rewarded by a drop of water. Odor C was never associated with outcomes. To isolate cue- and water-related signals from potential motion artifacts in recording and motor-related activity, mice were required to stay in an odor port for at least 1 s, and then to stay in a water port for 1 s to get water reward. The inter-trial-interval was fixed at 7 s after water onset in correct trials and at 9 s after any types of an error including violation of the stay requirement, no choice within 5 s after odor port out, and multiple pokes of an odor port after odor delivery. 1-Butanol, eugenol and cymene were diluted in 1/10 with mineral oil and randomly assigned to odor A, B, or C across animals. The odor-port assignment (left or right) was held constant in a single animal.

Mice were first trained only with pure odors and with the same amounts of water reward (~6 μl). After mice achieved greater than 90% accuracy, mice received a surgery for viral injection and fiber implantation. Following a 1 week recovery period, mice received re-training and then, mixtures of odor A and B (100/0, 90/10, 65/35, 35/65, 10/90, 0/100) were gradually introduced. After the accuracy of all the mixture odors achieved more than 50%, neuronal recording with fiber fluorometry was performed for five sessions. Subsequently, a task with different amounts of water was introduced. Mixtures of odor A and B (100/0, 65/35, 35/65, 0/100) but no odor C were used in this task. Each recording session started with 88–120 trials with an equal amount of water (~6 μl, the standard amount) in the first block to calibrate any potential bias on the day. In the second block, different amounts of reward were delivered in each water port. In order to make the water amounts unpredictable, one water port delivered big or medium size of water (2.2 and 0.8 times of the standard,~13.2 and 4.8 μl, BIG side) in a pseudo-random order, and another water port delivered medium or small size of water (0.8 and 0.2 times of the standard,~4.8 and 1.2 μl, SMALL side) in a pseudo-random order. Block 2 continued for 200 trials or until the end of recording sessions, whichever came earlier. A mouse performed 134.3 ± 3.4 (mean ± SEM) trials in block 2. The water condition (BIG or SMALL) was assigned to a left or right water port in a pseudo-random order across sessions. Recording was conducted for 40 min every other day to avoid potential bleaching. On days with no recording, animals were trained with pure odors A and B with the standard amount of water.

## Fiber photometry

Fiber fluorometry (photometry) was performed as previously reported (*Menegas et al., 2018*) with a few modifications. The optic fiber (400 μm diameter, Doric Lenses) allows chronic, stable, minimally disruptive access to deep brain regions and interfaces with a flexible patch cord (Doric Lenses, Canada) on the skull surface to simultaneously deliver excitation light (473 nm, Laserglow Technologies, Canada; 561 nm, Opto Engine LLC, UT) and collect GCaMP and tdTomato fluorescence emissions. Activity-dependent fluorescence emitted by cells in the vicinity of the implanted fiber's tip was spectrally separated from the excitation light using a dichroic, passed through a single band filter, and focused onto a photodetector connected to a current preamplifier (SR570, Stanford Research Systems, CA). During recording, optic fibers were connected to a magnetic patch cable (Doric Lenses,

MFP_400/430) which delivered excitation light (473 and 561 nm) and collected all emitted light. The emitted light was subsequently filtered using a 493/574 nm beam-splitter (Semrock, NY) followed by a 500 ± 20 nm (Chroma, VT) and 661 ± 20 nm (Semrock, NY) bandpass filters and collected by a photodetector (FDS10 × 10 silicone photodiode, Thorlabs, NJ) connected to a current preamplifier (SR570, Stanford Research Systems, CA). This preamplifier output a voltage signal which was collected by a NIDAQ board (National Instruments, TX) and Labview software (National Instruments, TX).

Calcium transients may not reflect spike counts, because of autofluorescence, bleaching, motion artifacts and inevitable normalization. We recorded tdTomato signals to monitor motion artifacts because a previous study showed that the red signals reflect motion artifacts reliably (*Matias et al., 2017*). Although we only applied this method in this study, additional methods using activity-independent wavelength of excitation (*Kudo et al., 1992*; *Lerner et al., 2015*) or examination of emission spectrum (*Cui et al., 2013*) may improve fidelity.

## Histology

Mice were perfused using 4% paraformaldehyde and then brains were sliced into 100 µm thick coronal sections using a vibratome and stored in PBS. Slices were then mounted in anti-fade solution (VECTASHIELD anti-fade mounting medium, H-1200, Vector Laboratories, CA) and imaged using a Zeiss Axio Scan Z1 slide scanner fluorescence microscope (Zeiss, Germany).

## Behavior analysis

We fitted % of odor mixture (X) to % of choice left or choice BIG ($\mu$) using generalized linear model with logit link function in each animal as previously reported (*Uchida and Mainen, 2003*).

$$\log(\mu/(1-\mu)) = Xb_1 + b_0$$

We first fitted a control block (block 1) and a reward-manipulation block (block 2) separately to examine difference of a slope, $b_1$ and a bias, $50-b_0/b_1$ of the curve. Next, to quantify shift of choice bias, we fitted choice of block 1 and block two together with a fixed slope, by fitting odor ($X_1$) and a block type ($X_2 = 0$ for block 1, $X_2 = 1$ for block 2) to choice.

$$\log(\mu/(1-\mu)) = X_1b_1 + X_2b_2 + b_0$$

Choice bias in block two was quantified choice bias as a lateral shift of the psychometric curve equivalent to % mixture of odors, $50 - (b_0 + b_2)/b_1$, which is a lateral shift compared to no bias, and $b_0/b_1 - (b_0 + b_2)/b_1$, which is a lateral shift compared to choice in block 1.

## GCaMP detection and analysis

To synchronize behavioral events and fluorometry signals, TTL signals were sent every 10 s from a computer that was used to control and record task events using Labview, to a NIDAQ board that collects fluorometry voltage signals. GCaMP and tdTom signals were collected as voltage measurements from current preamplifiers. Green and red signals were cleaned by removing 60 Hz noise with bandstop FIR filter 58–62 Hz and smoothing with moving average of signals in 50 ms. The global change within a session was normalized using a moving median of 100 s. Then, the correlation between green and red signals during ITI was examined by linear regression. If the correlation is significant ($p<0.05$), fitted tdTom signals were subtracted from green signals.

Responses were calculated by subtracting the average baseline activity from the average activity of the target window. Unless specified otherwise, odor responses were calculated by averaging activity from 1 to 0 s before odor port out (before choice) minus the average activity from the baseline period (1–0.2 s before odor onset). Responses after choice were calculated by averaging activity from 0 to 1 s after water port in minus the same baseline. Outcome responses were calculated by averaging activity from 0 to 1 s after water onset minus the same baseline. When comparing activity before and after water onset, average activity in 1–0.2 s before water onset was used as baseline. To normalize GCaMP signals across sessions within an animal, GCaMP signals were divided by average of peak responses during 1 s after odor onset in all the successful trials in the session. Z-scores of the signals were obtained using mean and standard deviation of signals in all the choice trials (from 2 s before odor onset to 6 s after odor onset) in each animal.

We built a regularized linear regression to fit cosine kernels (*Park et al., 2014*) (width of 200 ms, interval of 40 ms) to the activity of dopamine axons in each animal. We used down-sampled (every 20 ms) responses in all valid choice trials (trials with >1 s odor sampling time and any choice, −1 to 7 s from odor onset) for the model fitting. We used four different time points to lock kernels: odor onset ('odor'), odor port out ('movement'), water port in ('choice'), and water onset ('water'). Odor kernels consist of 4 types of kernels: 'base' kernels to span −960 to 200 ms from odor onset in all trials, and 'pure big' kernels in trials with a pure odor associated with big/medium water, 'pure small' kernels in trials with a pure odor associated with medium/small water, and 'mixture' kernels in trials with a mixture odor to span 0–1600 ms from odor onset. Movement kernels consist of 2 types of kernels: 'contra turn' kernels in trials with choice contra-lateral to the recording site, and 'ipsi turn' kernels in trials with choice ipsi-lateral to the recording site to span −1000 to 1200 ms from when a mouse exited an odor port. Choice kernels consist of 3 types of kernels: 'correct big' kernels in trials with correct choice of medium/small water and 'correct small' kernels in trials with correct choice of medium/small water to span −400 to 1200 ms from when a mouse entered a water port (water port in), and 'error' kernels in trials with choice error to span −400 to 5200 ms from water port in. Water kernels consist of 4 types of kernels: 'big water' kernels for big size of water, 'medium water big side' kernels for medium size of water at a water port of big/medium water, 'medium water small side' kernels for medium size of water at a water port of medium/small water, and 'small water' for small size of water to span 0–4200 ms after water onset. All the kernels were fitted to responses using linear regression with Lasso regularization with 10-fold cross validation. Regularization coefficient lambda was chosen so that cross-validation error is minimum plus one standard deviation. % explained by a model was expressed as reduction of a variance in the residual responses compared to the original responses. Contribution of each component in the model was measured by reduction of a deviance compared to a reduced model excluding the component.

We estimated response function to water in dopamine axons with linear regression with power function in each animal.

$$r = k(R^\alpha + c1 * S + c2)$$

where r is the dopamine axon response to water, R is the water amount, S is SMALL side (S = 1 when water was delivered at SMALL side, S = 0 otherwise). There are four different conditions, responses to big and medium water at a port of BIG side, and to medium and small water at a port of SMALL side. We first optimized α by minimizing average of residual sum of squares for each animal and then applied α = 0.7 for all the animals to obtain other parameters, k, c1, and c2. The response function was drawn with R as x-axis and r as y-axis. The amount of water to which dopamine axons do not respond under expectation of BIG or SMALL water was estimated by getting a crossing point of the obtained response function where the value is 0 (a zero-crossing point). The distribution of zero-crossing points was examined by linear regression of zero-crossing values against anatomical locations (anterior-posterior, dorsal-ventral, and medial-lateral). To visualize zero-crossing points on the atlas, zero-crossing values were fitted against anatomical locations with interaction terms using linear regression with elastic net regularization (α = 0.1) with 3-fold cross validation. The constructed map was sliced at a coronal plane Bregma +0.7 and overlaid on an atlas (*Paxinos and Franklin, 2019*).

To visualize activity pattern in multiple time windows at the same time, we stretched activity in each trial to standard windows. Standard windows from odor onset to odor poke out, and from odor poke out to water poke in, were determined by median reaction time and median movement time for each animal. For average plots of multiple animals, windows were determined by the average of median reaction times and of median movement times in all animals. The number of 100 ms bins in each time window was determined by dividing median reaction time and median movement time by 100. Dopamine responses in the window were divided into the bin number and the average response in each bin was stretched to 100 ms. The stretched activity patterns were used only for visualization, and all the statistical analyses were performed using original responses.

## Estimation of state values and TD errors using simulations

Matlab code for *Figure 7* is available at a source file 1. To examine how the value and RPE may change within a trial, we employed a Monte-Carlo approach to simulate animal's choices at a steady

state (i.e. after the animal learned the task). We used a Monte-Caro approach to obtain the *ground truth* state values as the animal progresses through task events without assuming a specific learning algorithm, under the assumption that the animal has learned the task. After obtaining the state values, we computed TD errors over the obtained state values.

## Model architecture

We considered two types of models. The variability and errors in choice in psychophysical performance can arise from at least two noise sources; noise in the variability in the process of estimating sensory inputs (perceptual noise) and noise in the process of selecting an action (decision noise). The first model contained only perceptual noise (*Green and Swets, 1966*), and the second model contained both perceptual and decision noise.

These models had different 'states' considering $N_S$ subjective odors ($N_S$ = 60 or four discrete states), choice (BIG versus SMALL), and different timing (inter-trial interval, odor port entry, odor presentation, choice, water port in, waiting for reward, and receiving feedback/outcome) (circles in *Figure 7A*).

We assumed $N_S$ possible subjective odor states (O') which comprise SubOdor1 and SubOdor2 states. We assumed that, in each trial, an internal estimate of the stimulus or a 'subjective odor' (O') was obtained by adding a noise to the presented odor stimulus (O) (one of the 4 mixtures of Odor A and B; 100/0, 65/35, 35/65, 0/100) (*Figure 7A–C*). In the model, the probability of falling on a given subjective odor state (O') is calculated using a Gaussian distribution centering on the presented odor (O) with the standard deviation, $\sigma$. We considered two successive states for subjective odor states in order to reflect a relatively long duration before an odor port exit.

As in the behavioral paradigm, whether the model receives a reward or not was determined solely by whether the presented odor (O) instructed the BIG side or SMALL side. Each subjective odor state contains cases when the presented odor (O) is consistent or congruent with the subjective odor (O'). For each subjective odor state, the probability of receiving a reward after choosing the BIG side, $p(BIG\ is\ correct) = f_B$, can be calculated as the fraction of cases when the presented odors instructed the BIG side. Conversely, the probability of reward after choosing the SMALL side is $p(SMALL\ is\ correct) = f_S = 1 - f_B$. Note that neither $f_B$ nor $f_S$ depends on reward size manipulations (as will be discussed later, the animal's choices will be dependent on reward size manipulations).

## Action selection

For each subjective odor, the model chose either the BIG or the SMALL side based on the value of choosing the BIG or SMALL side ($V_B$ and $V_S$ respectively, equivalent to the state value of the next state after committing to choose the BIG or SMALL side; see below for how $V_B$ and $V_S$ were obtained). In the first model which contains only perceptual noise, the side that is associated with a larger value is chosen. In the second model which contains both perceptual and decision noise, a choice is made by transforming $V_B$ and $V_S$ into the probability of choosing a given option using a sigmoidal function (e.g. Boltzmann distribution) (*Sutton and Barto, 1998*). In the softmax, the probabilities of choosing the BIG and SMALL side ($P_B$, $P_S$) are given, respectively, by,

$$P_B = \frac{e^{(V_B/(V_B+V_S))/\tau}}{e^{(V_B/(V_B+V_S))/\tau} + e^{(V_S/(V_B+V_S))/\tau}}$$

$$P_S = 1 - P_B$$

We also tested other stochastic decision rules such as Herrnstein's matching law (*Herrnstein, 1961*) or ε-greedy exploration (randomly selecting an action in a certain fraction [$\epsilon$] of trials) (*Sutton and Barto, 1998*). In Herrnstein's matching law, the probability of choosing the BIG side is given by,

$$P_B = \frac{V_S}{V_S + V_B}$$

The perceptual noise and a set of decision rule determine the behavioral performance of the model. The first model has only one free parameter, $\sigma$. The second model has one or no additional parameter ($\tau$ for softmax, or $\varepsilon$, for ε-greedy; no additional parameter for matching). We first

obtained the best fit parameter(s) based on the behavioral performance of all animals (the average performance in Block 2; that is *Figure 1C*, orange) by minimizing the mean squared errors in the psychometric curves.

For the first model, the best fit $\sigma$ was 21% Odor. We also tested with $\sigma$ of 5%, and the TD error dynamic was qualitatively similar. For the second model using the softmax rule, the best fit $\tau$ was 0.22 while $\sigma$ was 18% Odor.

## State values

The state value for each state was obtained as the weighted sum of expected values of available options which was computed by multiplying expected values of the option with probability of an option in the next step.

Outcome2 state represents the timing when the animal recognizes the amount of water. The state value is given by the amount of water that the model received (big, medium, small),

$$V_b = 2.2^\alpha$$

$$V_m = 0.8^\alpha$$

$$V_s = 0.2^\alpha$$

where the exponent $\alpha = 0.7$ makes the value function a concave function of reward amounts, similar to the fitting analysis of the fluorometry data (*Figure 4C*). Using $\alpha = 1$ (i.e. a linear function) did not change the results.

Ourcome1 state, or Water/No-water states (W and N, respectively) represent when the animal noticed the presence or absence of reward, respectively, but not the amount of reward. The value of a W (Water) state was defined by the average value of the next states. At the BIG side,

$$V_{WB} = (V_b + V_m)/2$$

whereas at the SMALL side,

$$V_{WS} = (V_m + V_s)/2$$

The values of N (No-water) states at the BIG and SMALL side are zero,

$$V_{NB} = 0$$

$$V_{NS} = 0$$

WaterPort1 and WaterPort2 states represent when the animal entered and stayed in the water port, respectively. The state value was obtained separately for the BIG and SMALL side. The value of choosing the BIG and SMALL sides is given by weighted sum of the values of the next states ($V_{WB}$, $V_{NB}$, $V_{WS}$, $V_{NS}$). The probabilities of transiting to the W and N states are given by the probability of receiving a reward given the choice (BIG or SMALL). As discussed above, these probabilities are given by $f_B$ and $f_S$, respectively. Thus,

$$V_B = f_B \cdot V_{WB}$$

$$V_S = f_S \cdot V_{WS}$$

We considered two successive states for WaterPort states to reflect a relatively long duration before receiving feedback/outcome. The two successive states had the same state values.

SubOdor1 and SubOdor2 states represent when the animal obtained a subjective odor (O') and before making a choice. The model chooses the BIG or SMALL side with the probability of $P_B$ and $P_S$, respectively, as defined above. Therefore, the state value of SubOdor1 and SubOdor2 was defined by the weighted sum of the values of the next states ($V_B$ and $V_S$),

$$V_{O'} = P_B V_B + P_S V_S$$

The two successive states had the same state values.

OdorOn state represents when the animal recognized the presentation of an odor but before recognizing the identity of that odor. The state value of the OdorOn state is defined by the weighted sum of the values of the next states (SubOdor1).

ITI state represents when the animal is in the inter-trial interval (i.e. before odor presentation). The value of ITI state was set to zero.

### TD errors

After obtaining state values at each state, we then computed TD errors using a standard definition of TD error which is the difference between the state values at consecutive time points plus received rewards at each time step (*Sutton, 1987*). For simplicity, a discounting factor was set to 1 (no discounting).

### Invalid trials

We also tested the effect of including invalid trials. At water acquisition, we included failures (20% of trials, value 0) where a mouse did not fulfil the requirement of odor poke duration (short odor poke), but did indicate a choice. At an odor port, failures resulted from multiple pokes of odor port (4% of trials), and a short odor poke (14% of trials). Values for these failures were set to 0. Existence or omission of these failures in models did not change the conclusion.

## Examination of correlation with models

To examine which model explains actual data better, we examined Pearson's correlation between model values and actual recording signals using bootstrapping. We tested models with deterministic choice and softmax choice, focusing on activity before and after choice. First, we randomly sampled 22 actual data (average activity before choice in each animal) in each trial type (eight trial types: BIG or SMALL choice, easy or difficult odor, correct or error) before and after choice. Because not all the animals have all the trial types, this sampling rule weighted to reflect rare trials equally well. Next, correlation of the sampled data ($22 \times 8 \times 2$ data) with each model was examined by Pearson's correlation, and difference of the correlation was calculated. The same procedure was repeated 500 times, and probability that correlation with a softmax choice model was equal to or smaller than a deterministic choice model was used as p-value.

## Mechanistic models with different ratios of inputs

To examine effects of different ratios of inputs whose efficacy is slightly different each other, we constructed a simplest model with only two inputs. Both inputs (excitatory and inhibitory) encode intact TD errors, but they mainly send information with excitation (10 times efficacy than inhibition). To vary ratios of inputs, we examined postsynaptic neurons that receive excitatory and inhibitory inputs at the ratio of 1:1 (balanced), 2:1 (more excitation) or 1:2 (more inhibition). Original TD errors in inputs were computed similar to *Figure 7*, except that water values are not fully learned so that expectation of water value is the average of expected water at the chosen water port BIG or SMALL and trained mount of water, to mimic dopamine axon responses to water (see above for model structure).

$$V_b = 2.2^\alpha$$

$$V_m = 0.8^\alpha$$

$$V_s = 0.2^\alpha$$

$$V_{trained} = 1^\alpha$$

$$V_{WB} = (V_b + V_m \ + \ 2V_{trained})/4$$

$$V_{WS} = (V_m + V_s + 2V_{trained})/4$$

## Video recording and analyses

A camera (BFLY-U3-03S2M, Point Grey Research Blackfly) was set on the ceiling of the behavioral box. We used infrared (IR) light (850 nm wave length, C and M Vision Technologies Inc, TX) to illuminate the arena and recorded video at 60 to 84 frames per s (fps) with H.264 video compression and streaming recording mode. The video was captured using the FlyCap2 software that accompanies the camera and processed using DeepLabCut (DLC) (*Mathis et al., 2018*). Frames were extracted for labeling using the k-means clustering algorithm, a process which reflects the diversity of images in each video. The resulting training dataset consisted of 400 video frames (40 frames per video) from 10 animals, seven animals with fluorometry fiber and three animals with no fiber, all of which were recorded while mice performed an odor-discrimination task in the same setup as this study. These manually labeled images were then used to refine the weights of a standard pretrained network (ResNet-50) for 1030000 training iterations. The network was designed to detect six body parts: a nose, left and right ears, a tail stem, a tail midpoint and a tail tip. 95% of the manually labeled frames were used to train DeepLabCut network, and 5% was used for evaluation of the network. The trained network was evaluated by computing the mean average Euclidean error (MAE) between the manual labels and the ones predicted by DeepLabCut. We trained three networks, using 3, 6 or 10 videos respectively. While MAE did not change across three trained networks (0.89 pixel, 0.88 pixel, 1.18 pixel for trained frames; 3.86 pixel, 2.59 pixel, 3.9 pixel for left-out frames), tracking performance dramatically improved with more training (see below), consistent with increase of likelihood estimation by DeepLabCut (0.87, 0.93, 0.97 for trained frames).

We processed 43 videos (two videos each from 21 animals and one video from a single animal). Video was synchronized with task events and fluorometry signals by sending TTL signals every 10 s from a computer that controlled the task, recorded mouse behavioral events to a fluorometry recording channel and at the same time turned on a 20 ms flash of infrared LED light (850 nm wave length, SparkFun Electronics, CO). To ensure that the LED light was invisible to a mouse, the light was set at the top edge of the behavioral box and covered by black light-shielding tape except for a small hole which was faced toward the camera. TTL-controlled light was detected in each video frame by thresholding the maximum intensity in the illuminated area.

Head position was obtained by averaging the positions of 3 body parts: the nose, left ear and right ear, and body position was obtained by averaging the positions of the head and tail stem. The optic fiber occasionally occluded or was mistaken for a body part, especially tail midpoint or tip. We therefore did not use locations of tail midpoint and tip in this study. To verify the tracking, tracked points were first visually examined by observing merged video with tracked points. Next, tracked points in adjacent frames were compared to detect biologically impossible gaps between frames. The warped points of >5.5 cm in a single frame decreased over training (1.6%, $8.0 \times 10^{-2}$% and $1.8 \times 10^{-3}$% respectively in an example video) and were filled by the average of tracking points in adjacent frames. To confirm tracking quality in the task, nose, head and body positions were examined during the time window when a mouse waited for water for 1 s. Locations of water ports were estimated at median of nose positions at 17 ms after water port entry in all the trials excluding trials with premature exit. Trials were excluded if nose positions were >2 cm, head positions were >2.5 cm or body positions were >8 cm from median of nose positions at 17 ms after water port entry in all the trials with >1 s in a water port. Excluded trials decreased over training (91.4%, 1.1%, 1.1%, respectively in an example video).

To examine fluorometry signals outside of the task, timepoints of movement start and stop were determined by transition from a quiet phase (<3 cm/s body speed) for >0.5 s to a moving phase (>3 cm/s body speed) for >0.5 s, and from a moving phase for >0.5 s to a quiet phase for >0.5 s, while a nose is >11 cm from an odor port side. To compare movement speed and fluorometry signals, 2000 frames were randomly picked in each video, and Pearson's correlation coefficient was obtained in each video.

To compare gross movement and fluorometry signals while animals were waiting for water in a water port, we calculated head and body distance traveled during the periods (50 frames, 833 ms) in each trial, and examined Pearson's correlation coefficient with average fluorometry signals during 0–1 s after water port entry. To examine whether movement is responsible for dopamine activity

modulation by accuracy after choice, fluorometry signals were linearly regressed with body speed and accuracy (correct or error) with elastic net regularization ($\alpha = 0.1$) with 5-fold cross validation.

## Randomization, blinding, and data exclusion

Photometry dataset was deposited at Dryad. No formal power analysis was carried out to determine the total animal number. We aimed for a sample size large enough to cover wide areas of the striatum. Chemicals were randomly assigned to an odor cue. Trial types (odors) were pseudorandomized in a block. Session types were pseudorandomized in a recording schedule. Animals were randomly assigned to a recording location. The experimenter did not know location of recording until the recording schedule was completed. No animals were excluded from the study: all analysis includes data from all animals. No trials were excluded from statistical analyses. To visualize average activity pattern in a stretched time window, outlier trials (maximum, minimum or average activity of a trial is outside of $3 \times$ standard deviation of maximum, minimum or average activity of all the trials) were excluded.

## Statistical analyses

Data analysis was performed using custom software written in MATLAB (MathWorks, Natick, MA, USA). All statistical tests were two-sided. For statistical comparisons of the mean, we used one-way ANOVA and two-sample Student's t tests, unless otherwise noted. Paired t tests were conducted when the same mouse's neural activity was being compared across different conditions or different time windows. The significance level was corrected for multiple comparisons using Holm–Sidak's tests unless otherwise indicated. All error bars in the figures are SEM. In boxplots, the edges of the boxes are the 25th and 75th percentiles (q1 and q3, respectively), and the whiskers extend to the most extreme data points not considered outliers. Points are drawn as outliers if they are larger than q3+1.5×(q3-q1) or q1-1.5×(q3-q1). Individual data points were overlaid on boxplots to compare striatal areas.

# Acknowledgements

We thank Ju Tian, William Menegas, HyungGoo Kim, Takahiro Yamaguchi, Yu Xie and Alexander Mathis for technical assistance, Kristen Fang, Grace Chang and Sakura Ikeda for assistance in animal training and histology, and Adam Lowet, Sara Pinto dos Santos Matias, Michael Bukwich, Malcolm Campbell, Paul Masset, and all lab members for discussion. We also thank V Jayaraman, R Kerr, D Kim, L Looper, and K Svoboda from the GENIE Project, Janelia Farm Research Campus, Howard Hughes Medical Institute for AAV-FLEX-GCaMP7f. This work was supported by National Institute of Mental Health R01MH095953, R01MH101207, R01MH110404, R01NS108740 (NU); and Japan Society for the Promotion of Science, Japan Science and Technology Agency (HM, ITK).

# Additional information

### Competing interests

Naoshige Uchida: Reviewing editor, *eLife*. The other authors declare that no competing interests exist.

### Funding

| Funder | Grant reference number | Author |
| --- | --- | --- |
| Japan Society for the Promotion of Science | | Iku Tsutsui-Kimura Hideyuki Matsumoto |
| National Institute of Mental Health | R01MH095953 | Naoshige Uchida |
| National Institute of Mental Health | R01MH101207 | Naoshige Uchida |
| National Institute of Mental Health | R01MH110404 | Naoshige Uchida |

| National Institute of Mental Health | R01NS108740 | Naoshige Uchida |

The funders had no role in study design, data collection and interpretation, or the decision to submit the work for publication.

## Author contributions

Iku Tsutsui-Kimura, Conceptualization, Funding acquisition, Investigation, Writing - original draft, Writing - review and editing; Hideyuki Matsumoto, Conceptualization, Funding acquisition, Writing - review and editing; Korleki Akiti, Formal analysis, Methodology, Performed video analyses; Melissa M Yamada, Methodology, Writing - review and editing, Performed video analyses; Naoshige Uchida, Conceptualization, Funding acquisition, Writing - original draft, Writing - review and editing; Mitsuko Watabe-Uchida, Conceptualization, Formal analysis, Supervision, Investigation, Writing - original draft, Writing - review and editing

## Author ORCIDs

Iku Tsutsui-Kimura (iD) https://orcid.org/0000-0001-7554-4764
Naoshige Uchida (iD) http://orcid.org/0000-0002-5755-9409
Mitsuko Watabe-Uchida (iD) https://orcid.org/0000-0001-7864-754X

## Ethics

Animal experimentation: All procedures were performed in accordance with the National Institutes of Health Guide for the Care and Use of Laboratory Animals and approved by the Harvard Animal Care and Use Committee (protocol #26-03). All surgeries were performed under aseptic conditions with animals anesthetized with isoflurane (1-2% at 0.5-1.0 l/min). Analgesia was administered pre (buprenorphine, 0.1 mg/kg, I.P) and postoperatively (ketoprofen, 5 mg/kg, I.P).

## Decision letter and Author response

Decision letter https://doi.org/10.7554/eLife.62390.sa1
Author response https://doi.org/10.7554/eLife.62390.sa2

# Additional files

## Supplementary files

• Source code 1. MATLAB code to model state value and TD error. This code was used to generate *Figure 7B–G*, and *Figure 7—figure supplement 1*.

• Transparent reporting form

## Data availability

A source code file has been provided for Figure 7. Fluorometry data has been deposited in Dryad available at: https://doi.org/10.5061/dryad.pg4f4qrmf.

The following dataset was generated:

| Author(s) | Year | Dataset title | Dataset URL | Database and Identifier |
| --- | --- | --- | --- | --- |
| Tsutsui-Kimura I, Matsumoto H, Uchida N, Watabe-Uchida M | 2020 | Dopamine axon population Ca signals in the striatum during odor cue- and reward-based choice tasks in mice | http://dx.doi.org/10.5061/dryad.pg4f4qrmf | Dryad Digital Repository, 10.5061/dryad.pg4f4qrmf |

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
