## [Decision Letter]

**Acceptance summary:**

In this paper, fiber photometry and a perceptual choice task are utilized to provide a very interesting and important window onto the activity of dopamine axons in three different regions of the striatum. Unexpectedly, activity in all three regions was found to be quite similar, in contrast to accepted function of dopamine actions in different striatal. In addition, a model is provided that captures some features of the behavior and may explain this homogeneity.

**Decision letter after peer review:**

Thank you for submitting your article "Distinct temporal difference error signals in dopamine axons in three regions of the striatum in a decision-making task" for consideration by *eLife*. Your article has been reviewed by three peer reviewers, and the evaluation has been overseen by a Reviewing Editor and Kate Wassum as the Senior Editor. The following individual involved in review of your submission has agreed to reveal their identity: Melissa J Sharpe (Reviewer #4).

The reviewers have discussed the reviews with one another and the Reviewing Editor has drafted this decision to help you prepare a revised submission.

As the editors have judged that your manuscript is of interest, but as described below that additional analyses and major text revisions are required before it is published, we would like to draw your attention to changes in our revision policy that we have made in response to COVID-19 (https://elifesciences.org/articles/57162). First, because many researchers have temporarily lost access to the labs, we will give authors as much time as they need to submit revised manuscripts. We are also offering, if you choose, to post the manuscript to bioRxiv (if it is not already there) along with this decision letter and a formal designation that the manuscript is "in revision at *eLife*". Please let us know if you would like to pursue this option. (If your work is more suitable for medRxiv, you will need to post the preprint yourself, as the mechanisms for us to do so are still in development.)

Summary:

In this paper, the authors use fiber photometry coupled to a perceptual choice task to provide a functional window onto the activity of dopamine axons in three different regions of the striatum (VS: the ventral striatum; DMS: the dorso-medial striatum; and DLS: the dorso-lateral striatum). Unexpectedly, fluorescence signals in all regions are exhibit only relatively subtle differences of baseline (DLS has an elevated baseline) and movement sensitivity (largely in the DMS). The authors lay out a picture of the various signals, and also provide a model that captures some aspects of the behavior.

Essential revisions:

1) While the models used to explain the data are quite elegant, the overall results remain correlative. There are no experimental manipulations to test the primary conclusions. Most notably, the lack of "dopamine dip" in the DLS following a smaller than expected reward may support a role for this region in habitual behaviors. However, this remains speculative, as there is no operationally defined metric that animals are unambiguously showing habits.

2) The homogenous pattern of dopamine neuronal signaling across striatal regions during reward-directed behavior in the current study contrasts starkly with extensive heterogeneity observed in prior studies using Ca^2+^ imaging (Howe et al., 2016; Parker et al., 2016) or voltammetry (Brown et al., 2011; Willuhn et al., 2012 PNAS). For example, Oyama et al., 2010, record single-unit activity in DLS and find that these neurons do not show negative errors in response to unexpected omission of reward in a similar task used here. Please provide a potential and detailed explanation for why such dramatic differences are observed in the current work, that cannot be solely related to differences in the technique used.

3) The idea that the DLS might be engaged in something like the actor learning part of the actor-critic learning rule (rather than Q learning, say) has been relatively widely suggested – the authors might comment on the observation in REINFORCE rules (that the paper cites) that it is possible to add an "arbitrary" action-independent baseline to the equivalent of the prediction error and derive the same expected change in the weights (only the variance is affected). Does this not imply that the baseline shift in the DLS is unlikely to be associated with a different functional property?

4) It would be expected that the model would make a much larger distinction between the medium reward in the BIG than the SMALL condition than was apparent in the data. Indeed, one of the puzzles for me about the paper was the tiny effect of the BIG vs SMALL side (as in Figure 4C, for instance) – why might that be? Would the authors' previous results not predict a larger difference?

5) The FP data shows three peaks in DA in correct trials, whereas the model only really shows two. The second peak, associated with leaving the sampling well is important – particularly for the DMS – could the authors comment on its absence in the model?

6) Further analysis is required from the transition between block 1 and block 2 and the speed of adaptation could provide useful cues about the employment of goal-directed or habitual values, and distinguish DMS and DLS.

7) It is unclear why the data is combined across regions in showing responses to the odor or choice. Is the only time these signal diverge during reward delivery? I think it would significantly increase the novelty and impact of the findings if the authors could focus on representing the differences between these regions in their figures (e.g. Figure 2 and 3), and conduct the analyses in relation to the different regions (rather than clumping all regions in the modeling). What are the parameters that change when data is sorted out this way? The claim that these data all line up a TD error simply cannot be the case, because DLS does not show the hallmark dip in activity to an unexpected reward that would be expected of a TD error.

8) The claim is made that DLS activity is consistent with the "law of exercise" proposed by Thorndike, encoding how to perform motor sequences. However, these data are also consistent with this region encoding the reward outcome as a stimulus (i.e. the smallest reward is encoded as a small reward, instead of a reduction in value). Alternatively, it could be encoding the "long run history" of rewards (and impervious to single reductions in reward magnitude), which is consistent with current theories of habitual learning. Therefore, suggesting changes to existing theories on the basis of a single characteristic of neural activity in a limited data set, without a behavioral test that would facilitate such an understanding is not advisable. In fact, this can be removed with no detrimental effect to the paper.

9) The fact that these data can be modeled in a TD framework does not mean dopamine terminals are acting as a model-free TD error. This is because there is no test outside this hypothesis. There is nothing other than reward "value" that is manipulated. And dopamine neurons are capable of acting outside a model-free TD error. This has been demonstrated many times by many different labs in the past 3-5 years; all of this should be noted and discussed. Indeed, dopamine neurons in VTA respond to unexpected changes in reward identity (Takahashi et al., 2017), and stimulation of dopamine neurons drives learning between neutral cues without endowing them with "model-free" value (e.g. Sharpe et al., 2020, Nature Communications). This is not predicted by a model-free TD signal. Indeed, similar "model-based" findings have been found in ventral striatum (e.g. Corbit and Balleine, 2011, Journal of Neuroscience; Cerri et al., 2014, Behavioral Neuroscience).

10) Please rephrase your interpretation as follows: " our modeling approach is consistent with activity at DA terminals functioning within the lines of a TD error framework in this particular reward-learning task, while we acknowledge that it is likely these signals are also acting outside of this framework in other tasks".

11) It is crucial to report whether animals engage in licking in the water (or sample) port before the reward is provided? Does this covary with model 'value' and confidence? Does it covary with FP signals?

[Editors' note: further revisions were suggested prior to acceptance, as described below.]

Thank you for resubmitting your work entitled "Distinct temporal difference error signals in dopamine axons in three regions of the striatum in a decision-making task" for further consideration by *eLife*. Your revised article has been evaluated by Kate Wassum (Senior Editor) and a Reviewing Editor.

The manuscript has been improved but there are some remaining issues that need to be addressed before acceptance, as outlined below.

In subsection “Similarity of dopamine axon signals across the striatum” paragraph three, optimism in Q learning should not be described without qualification. The fact that Q learning uses max_b Q(s(t+1),b) at the next state (s(t+1)) is not thought of being borne of optimism – this in fact reflects what the subject thinks it could do at that state (although it would indeed be optimistic to think that it would always do that). Confusion may arise in that optimistic Q values are associated with the motivation to explore. If the authors wish to include the assertion about Q learning, they should expand on their exact point.

The" dip" in the signal is important for the TD error. And if the proxy for the DLS signals is "more excitation" represented in Figure 8, this should still predict a dip in the DLS (albeit a smaller one). However, signals in DLS still show an increase in activity from baseline during water omission. This should be clarified.

The authors propose that the observed signals look like the scalar difference between the expected an observed value (with the exception of the DLS), but are not necessarily used to attach this scalar value product to the antecedent cue or action. This should be specifically and explicitly stated in the text as it is not necessarily scalar value that attaches to antecedent events, which is alluded to by referencing and modeling using the model-free TD algorithm.

Please include summary statistics in additional to P values where appropriate (e.g., for ANOVA and t-tests).

---

## [Author Response]

Essential revisions:1) While the models used to explain the data are quite elegant, the overall results remain correlative. There are no experimental manipulations to test the primary conclusions. Most notably, the lack of "dopamine dip" in the DLS following a smaller than expected reward may support a role for this region in habitual behaviors. However, this remains speculative, as there is no operationally defined metric that animals are unambiguously showing habits.

Thank you for your comment. It is important to distinguish what we can conclude directly from the data and speculations. The idea that the lack of dopamine dip in the DLS supports a role in habit is our speculation, given the literature pointing to the role of DLS in habitual behaviors and our present results. Before addressing the concern, let us explain the main goal of the study and the main conclusions that we can draw directly from the data.

We designed our task to quantitatively examine effects of reward value, prediction error and confidence separately on dopamine axon activity. From observation in these experiments, we drew several conclusions.

Our primary conclusions consist of three points;

1) Dopamine axon activities in VS, DMS and DLS are very similar and consistent with RPE (TD error) during value- and perception-based decision task.

2) Excitation thresholds of dopamine reward responses are different across areas.

3) Representation of confidence- or accuracy-like signals can naturally emerge as dynamics of TD errors, providing a unified account on dynamic dopamine signals in the task.

We believe that each of these findings is important, which trigger many types of future studies. One of future directions we propose is the relationship between positively-biased RPE signals in dopamine axons in DLS that we found, and function of DLS in habit/skill that had been proposed in previous many studies. This is what we discussed in Discussion, but never in Results. In order to distinguish our conclusions in the Results and Discussion, we added sentences to summarize our main conclusions.

“We present three main conclusions. First, contrary to the current emphasis on diversity in dopamine signals (and therefore, to our surprise), we found that dopamine axon activity in all of the three areas exhibited similar dynamics. […] Nonetheless, dopamine axon signals overall exhibited temporal dynamics that are predicted by TD errors, yet, the representation of TD errors was biased depending on striatal areas.”

Our task is not designed specifically to test habitual behaviors, and the ability of the mice to bias their choices according to the reward amounts suggests that the mice were not behaving completely in a habitual mode (Figure 1). The stochastic variation of reward amounts might have facilitated this flexible behavior. Nonetheless, this result does not necessarily indicate that the brain area related to habit learning and its reinforcement signals are completely shut down. It is likely that whether the animal exhibits goal-directed versus habitual behaviors is determined by sensitive balance between various factors (e.g. relative strengths of activity in the brain areas controlling goal-directed and habit behaviors), and dopamine input to DLS might still be providing “permissive” reinforcement signals. Furthermore, the choice behavior (decision as to left versus right) is just one aspect of behaviors that the animal has to learn in our paradigm. The animal learns various task-related, sequence of movements through learning in the task (e.g. smooth movement from the center to a reward port). In this sense, we believe that it is still remarkable that dopamine signals in DLS possess properties that could readily explain aspects of habit or skill learning in the current task. Our focus is to extract a rule that governs dopamine axon activity in each striatal area by comparing activity in a single task. In any case, whether the behavior in our task are habitual or goal-directed would not affect the validity of our three main conclusions listed above.

Nonetheless, to reflect the limitation that the reviewers pointed out, we have deleted the word "habit" from Abstract, and replaced the concluding sentence with a more general conclusion as follows:

“The differences in dopamine signals in these regions put specific constraints on the properties of behaviors controlled by dopamine in these regions.”

We also agree that the Discussion in our previous version of the manuscript focused too much on this one point (habit/skill). Following the recommendations in other comments below, we have shortened our discussion on habit/skills, and more evenly discuss the three points in the revised manuscript.

2) The homogenous pattern of dopamine neuronal signaling across striatal regions during reward-directed behavior in the current study contrasts starkly with extensive heterogeneity observed in prior studies using Ca^2+^ imaging (Howe et al., 2016; Parker et al., 2016) or voltammetry (Brown et al., 2011; Willuhn et al. 2012 PNAS). For example, Oyama et al., 2010, record single-unit activity in DLS and find that these neurons do not show negative errors in response to unexpected omission of reward in a similar task used here. Please provide a potential and detailed explanation for why such dramatic differences are observed in the current work, that cannot be solely related to differences in the technique used.

This is a very important point, and we expanded Discussion related to previous studies in subsection “Similarity of dopamine axon signals across the striatum**”**

Briefly, our results were consistent with many previous studies. The most important difference between these and the present study is that many of the previous studies used PREDICTED reward to examine dopamine responses to reward. In contrast, we varied the reward amounts parametrically, and introduced some stochasticity in reward. We speculate that more homogeneous pattern that we observed is, at least in part, due to this task design.

To illustrate, we made an additional figure presenting the information in the way similar to the previous studies (Figure 3—figure supplement 1). As is clear in this new figure, if we presented dopamine axon activity in a similar manner as previous studies, the results are similar to the previous studies (these plots emphasize difference). In other words, if we only conducted the standard experiments with fixed amounts of water, our conclusion would have been the same as previous studies, emphasizing diversity rather than similarity. However, we performed additional experiments, and found that if the water amounts are manipulated in a probabilistic manner to cause quantitative RPE, RPE signal patterns looked similar with one another (Figure 4A). Thus, some of the diversity observed in previous studies might be caused by predictability of reward amounts, and could be explained by differences in excitation thresholds. We added a comment on similarity of results more clearly :

“Our observation of similarity across striatal areas (Figure 4A) would give an impression that these results are different from previous reports. […] Taken together, although we cannot exclude the possibility that dopamine activity in DLS is modulated by a specific movement in particular conditions, our results do not support that TD error-like activity in DLS is generated by a completely different mechanism or based on other types of information than other dopamine neuron populations.”

Our results in DMS are consistent with previous studies that reported low and somewhat mysterious responses to reward (Brown et al., 2011; Parker et al., 2016). We noticed that while animals were trained with fixed amounts of water, some of dopamine axon signals in DMS did not exhibit clear responses to water, and in average water responses were smaller than other areas. Once reward amounts became dynamic, dopamine axons in DMS showed clear responses according to RPE, similar to the previous observation that dopamine responses to reward in DMS emerged after contingency change (Brown et al., 2011). We state:

“Notably, the study (da Silva et al., 2018) also used predictable reward (fixed amounts of water with 100% contingency) to examine dopamine responses to reward.”

Why dopamine in DMS shows movement-related activity, and why it sometimes does not show responses to water are not fully addressed in this study. We added more discussion about each previous finding.

“As pointed out in a previous study (Lee et al., 2019), this movement-related activity in DMS is unlikely to be a part of RPE signals.”

“Why are reward responses in DMS sometimes observed and sometimes not? We found that the response function for water delivery in dopamine axons in different striatal areas showed different zero-crossing points, the boundary between excitatory and inhibitory responses (Figure 4). […] It will be important to further examine in what conditions these dopamine neurons lose responses to water, or whether there are dopamine neurons which do not respond to reward in any circumstances.”

We also emphasize the limitations of our photometry method in our study.

“We also have to point out that because fluorometry in our study only recorded average activity of dopamine axons, we likely missed diversity within dopamine axons in a given area.”

“However, because we only recorded population activity of dopamine axons, our results do not exclude the possibility that some dopamine neurons that do not respond to reward also project to DLS.”

Further, to compare our recording data with previous studies, we conducted video analyses using DeepLabCut, and compared movement and dopamine activity. Consistent with previous observation (Coddington and Dudman, 2018), we observed slight modulation of dopamine axon signals with movement. Now the new results are presented in Figure 2—figure supplement 1 and Figure 7—figure supplement 6.

3) The idea that the DLS might be engaged in something like the actor learning part of the actor-critic learning rule (rather than Q learning, say) has been relatively widely suggested – the authors might comment on the observation in REINFORCE rules (that the paper cites) that it is possible to add an "arbitrary" action-independent baseline to the equivalent of the prediction error and derive the same expected change in the weights (only the variance is affected). Does this not imply that the baseline shift in the DLS is unlikely to be associated with a different functional property?

Thank you for providing us with the opportunity to address this important and interesting point. We have expanded our Discussion on this. Briefly, we proposed preference learning in DLS (Sutton and Barto, 1998) instead of value learning, as proposed in actor-critic models, but we do not propose that DLS does policy gradient updates such as in REINFORCE because the underlying assumption that behaviors (or policy) converge to the optimal solution might not be consistent with the concept of habit. Previous studies indicated that DLS is responsible for habit and skill, even when it is not optimal. Instead, reinforcing rules could be (but are not limited to) direct updates as proposed in an original actor-critic model (Barto et al., 1983; Takahashi et al., 2008). This is different from the REINFORCE rules, which is inversely proportional to action probability.

“We propose that policy learning may be a better way to conceptualize the function of the DLS, as was proposed in previous studies (Sutton and Barto, 1998; Takahashi et al., 2008). Instead of finding an optimal solution seen in policy gradient methods (Sutton and Barto, 2018), positively-biased TD errors in DLS may directly reinforce taken actions as proposed originally (Barto et al., 1983; Sutton and Barto, 1998), thus preserving the law of effect but also emphasize the law of exercise.”

4) It would be expected that the model would make a much larger distinction between the medium reward in the BIG than the SMALL condition than was apparent in the data. Indeed, one of the puzzles for me about the paper was the tiny effect of the BIG vs SMALL side (as in Figure 4C, for instance) – why might that be? Would the authors' previous results not predict a larger difference?

This is a good point. In retrospect, this might be due to suboptimal conditions of the current design with respect to our ability to observe the effect of BIG and SMALL expectations on medium-sized reward. The range of reward is 0 to 2.2 L, whereas the expectations differ by less than the half of the range (1.0 L). Relative to the expectation, the expected positive and negative RPEs are -0.7 (medium reward in BIG) and +0.3 (medium reward in SMALL). In our typical experiments, reward responses are reduced by approximately the half of the response of the full reward response even when the reward is fully expected (Tian and Uchida, 2015; Eshel et al., 2015; Starkweather et al., 2017), likely due to temporal uncertainty of reward timing (Fiorillo et al., 2008; Kobayashi et al., 2008). Therefore, the relatively small variation (1.0 μL) in expectation might have not been the optimal condition to observe the effect of expectation although our point is that we can still see the significant difference between the conditions (BIG versus SMALL). Another reason why we did not see a big effect is relatively short training. Additionally, we cannot exclude the possibility that mice are generally incapable of distinguishing these relatively minor differences. In order to observe a larger difference, we would need to examine various task parameters and optimize the task condition in future experiments.

We added a Discussion in subsection “Limitations and future directions”

“Our task incorporated a typical perceptual task using different levels of sensory evidence (Rorie et al., 2010; Uchida and Mainen, 2003) into a value-based learning with probabilistic reward manipulation using 2 sets of different sizes of water. Although the task was demanding to mice, we were able to detect RPE natures in dopamine axon activity without over-training. However, the difference of prediction BIG versus SMALL sides was still small. Further, most analyses relied on pooled data across sessions because of the limited number of trials in each trial type, especially error trials. Further improvement of the task will facilitate more quantitative analyses over learning.”

5) The FP data shows three peaks in DA in correct trials, whereas the model only really shows two. The second peak, associated with leaving the sampling well is important – particularly for the DMS – could the authors comment on its absence in the model?

The second peak corresponds to the peak we identified in Figure 2C “Movement”. Nathaniel Daw devoted a whole *eLife* paper to address this question, and concluded that movement-related activity in DMS-projecting dopamine neurons are not a part of RPE. We did not see movement-related activity in TD errors in our models, consistent with this idea. We added a citation of this paper.

“Third, interestingly, however, our results showed consistent deviation from what TD model predicts. As reported previously (Parker et al., 2016), during choice movements, contra-lateral orienting movements caused a transient activation in the DMS, whereas this response was negligible in VS and DLS. As pointed out in a previous study (Lee et al., 2019), this movement-related activity in DMS is unlikely to be a part of RPE signals.”

6) Further analysis is required from the transition between block 1 and block 2 and the speed of adaptation could provide useful cues about the employment of goal-directed or habitual values, and distinguish DMS and DLS.

In this manuscript, we are proposing that recipients of dopamine signals (neurons in the striatum) potentially learn in a slightly different way, because excitation threshold of TD error is slightly different across striatal areas. Thus, neurons in DLS may learn habit, not DLS-projecting dopamine neurons, themselves. To distinguish roles of dopamine and the striatum in our proposals more clearly, we added an illustration in a new figure, Figure 8D.

The difference in learning speed in dopamine neurons is an interesting question and we should design tasks to parametrically test their speed. Our task has limitations as we addressed/discussed in the answer to question 4. We explicitly added a sentence about the learning:

“Further improvement of the task will facilitate more quantitative analyses over learning.”

Nevertheless, we compared effects of prediction BIG versus SMALL in dopamine axon responses to water as in Figure 4D, but separated trials in early and late sessions (trials 1-50, and after 100). We did not observe clear differences across striatal areas (p=0.99, ANOVA, difference between early and late across 3 striatal areas).

7) It is unclear why the data is combined across regions in showing responses to the odor or choice. Is the only time these signal diverge during reward delivery? I think it would significantly increase the novelty and impact of the findings if the authors could focus on representing the differences between these regions in their figures (e.g. Figure 2 and 3), and conduct the analyses in relation to the different regions (rather than clumping all regions in the modeling). What are the parameters that change when data is sorted out this way? The claim that these data all line up a TD error simply cannot be the case, because DLS does not show the hallmark dip in activity to an unexpected reward that would be expected of a TD error.

We observed different dopamine axon activities across the striatum not only during reward delivery but also at cue responses (Figure 5G-I, Figure 6—figure supplement 1). We apologize that we did not have average traces at different time windows as in Figure 4A in the previous version. We have added average traces in each area for both before choice and after choice in Figure 6—figure supplement 1.

Our main point is that other than the lack of “dip”, other responses are consistent with TD errors which correspond to changes in state values. Following the original definition of TD errors, the dip is just one aspect. To produce different zero-crossing points across the striatal areas, we propose 3 different possibilities. First, the difference might come from the same mechanism as the distributional RL (Dabney et al., 2020). Some dopamine neurons might be optimistic and some are pessimistic. The model is fully described in the published paper, and how to combine this idea and TD is a hot topic in the field. Second, DLS-projecting dopamine neurons may receive something additional whenever animals obtain a positive outcome as "success premium", although we do not have an anatomical clue for this mechanism. Third, which is not exclusive with the first idea, dopamine subpopulations with different projection targets may receive different ratios of inputs, some of which represent already partially RPE (Tian et al., 2016). In this previous study, we observed many presynaptic neurons with RPE, but RMTg uniquely showed a flipped version of RPE (the sign is opposite to dopamine RPE). Interestingly, inactivation of RMTg only affects negative RPE in dopamine neurons although RMTg show both negative and positive RPE (Li et al., 2019), suggesting that excitation of these neurons is more important. Further, RMTg shows a topographic projection pattern to dopamine neurons (Jhou et al., 2019). From these results, we propose that different ratios of inputs to dopamine neurons cause different zero-crossing points in dopamine neurons. To illustrate, we added a simple model with only two inputs in Figure 8A-C.

“Mechanistically, biases in dopamine signals may stem from a difference in the excitation-inhibition balance at the circuit level. […] It will be fascinating if we can connect all these levels of studies into functional meaning in the future.”

8) The claim is made that DLS activity is consistent with the "law of exercise" proposed by Thorndike, encoding how to perform motor sequences. However, these data are also consistent with this region encoding the reward outcome as a stimulus (i.e. the smallest reward is encoded as a small reward, instead of a reduction in value). Alternatively, it could be encoding the "long run history" of rewards (and impervious to single reductions in reward magnitude), which is consistent with current theories of habitual learning. Therefore, suggesting changes to existing theories on the basis of a single characteristic of neural activity in a limited data set, without a behavioral test that would facilitate such an understanding is not advisable. In fact, this can be removed with no detrimental effect to the paper.

There are several points we have to clarify here. First, although "a limited data set" was blamed here, our data is the first dataset which examined potential differences of teaching signals by dopamine in different striatal areas, utilizing parametric tasks. In other words, existing theories are not based on actual dopamine activity in DLS, nor did they propose that the lack of “dip” may be related to habit learning. We reiterate that although the current task is not designed specifically to test habitual behaviors, dopamine signals in DLS may still be providing permissive reinforcement signals. Second, the idea that DLS plays an important role in habit/skill is not our new proposal, but already supported by substantial evidence in the past, most notably by studies involving lesions and pharmacological manipulations in DLS. Our speculation that the lack of dip in DLS dopamine signals may underlie habit learning builds on this rich literature pointing the importance of DLS in habit and skill and the present result. We clarified these two points in Discussion.

We appreciate that the reviewers agree that our data is valid even without the discussion that we made. As we answered to question 1, we more emphasized our findings themselves, and deleted most of the discussion about sequential behaviors. We would like to respectfully argue, however, that our discussion on habit learning through DLS dopamine signals is a natural logical speculation that one can make based on the observed results and the existing literature. Our goal is to contextualize our observations in existing literature and make proposals for predictions, future directions and future experiments, which are one of the most important goals of Discussion section in original research paper, and our discussions are well-based on the data.

That habit learning follows "law of exercise" or repetition/training is not a new idea and well accepted in wide fields. In spite of this well-known phenomenon, current popular computational theories often use value-based, model-free reinforcement learning to model habit. To challenge these theories, several models were proposed recently, and one of these (Miller et al., 2019) uses action history as a teaching signal, which sometimes approximates long run history of rewards because animals tend to repeat rewarded actions. As Miller et al. emphasized, this way of computation allowed habit learning to follow law of exercise, although this computational model was purely theoretical, with no underlying neuronal mechanism. Our proposal fits well with the idea of learning from repetition. While Miller et al. totally removed effects of reward, our proposal is a fusion, based on observation of dopamine activity. Dopamine axon activity in DLS was sensitive to water amounts, and did not show excitation with mere action with no outcome. However, it showed strong excitation even with small reward. Thus, dopamine in DLS can function as a teaching signal emphasizing law of exercise, but not ignoring reward, or value-less.

Existing computational models have problems and that is why these recent studies are trying to revise models for habit. However, these new models do not have a neuronal basis. On the other hand, there is a history of manipulation studies that showed the importance of DLS and dopamine in habit/skill. The present study builds on these literatures and our finding in DLS dopamine signals bridges the gap between these behavioral and computation studies.

There are many existing theories to explain DLS function. One of them is reinforcement learning theory, assuming that dopamine activity is uniform across the striatum including DLS. Here, we designed a task to propose the potential function of dopamine in DLS in reinforcement learning, based on actual recording data. We discuss what positively-biased teaching signals can affect learning in downstream areas. An important point here is that learning with balanced-TD error can converge to value, whereas learning with biased TD error does not, with simple reinforcement learning. Specifically, positively-biased TD error would cause reinforcement by repetition. We believe that this is a natural logical speculation that one can draw based on both the observed data, and well-established reinforcement learning theory.

Now a totally different question is what can produce positively-biased TD error-like signals. Because this question pertains to the next step, we have a different section in Discussion subtitled “Potential mechanisms underlying diverse TD error signals”

As we discussed at question 7, we discuss more than three possibilities. Now, the possibility of sensory prediction error is additionally raised here. We have not seen any data showing that dopamine in DLS encodes sensory prediction error, nor an existing theory to connect sensory prediction error and habit. We showed previously that activity of TS-projecting dopamine neurons uniquely co-varies with stimulus intensity (Menegas et al., 2018), but TS is a distinct area from DLS. Another proposal here is long run history of reward in dopamine neurons. Single unit recording data indicated that dopamine neurons in the VTA encoded longer time-scale of reward value, compared to lateral SNc (Enomoto et al., 2020), in contrast to the reviewers' suggestion.

9) The fact that these data can be modeled in a TD framework does not mean dopamine terminals are acting as a model-free TD error. This is because there is no test outside this hypothesis. There is nothing other than reward "value" that is manipulated. And dopamine neurons are capable of acting outside a model-free TD error. This has been demonstrated many times by many different labs in the past 3-5 years; all of this should be noted and discussed. Indeed, dopamine neurons in VTA respond to unexpected changes in reward identity (Takahashi et al., 2017, Neuron), and stimulation of dopamine neurons drives learning between neutral cues without endowing them with "model-free" value (e.g. Sharpe et al., 2020, Nature Communications). This is not predicted by a model-free TD signal. Indeed, similar "model-based" findings have been found in ventral striatum (e.g. Corbit and Balleine, 2011, Journal of Neuroscience; Cerri et al., 2014, Behavioral Neuroscience).

Here we do not see any disagreement and we were somewhat puzzled why the reviewers raised this as a major issue. We completely agree with the reviewers that “the fact that these data can be modeled in a TD framework does not mean dopamine terminals are acting as a model-free TD error”. In the manuscript, we do not make any claim as to whether the observed dopamine signals are model-free or model-based. If our manuscript gave even slight impression that we are claiming that dopamine neurons only take into account model-free information, that would be our mistake and we would like to correct it. However, we would need to know which of our statements invoked such a concern before responding to it.

It is clear that, based on previous studies (which the reviewers have kindly listed but also Bromberg-Martin et al., 2010; Takahashi et al., 2011; Starkweather et al., 2017, 2018; Babayan et al., 2018; and perhaps Daw et al., 2011), dopamine neurons have access to model-based information or some form of inference using a model (or task states), and these conclusions were drawn by experiments designed to address this specific question. The present experiments are not designed for that, but to distinguish specific variables previously proposed to co-vary with dopamine signals such as RPE (TD error), sensory evidence, and reward values. Because computation of any of these variables may involve model-free or model-based mechanisms, our experiments have no power for distinguishing model-free versus model-based computations. Generally, we are reluctant to over-simplify dopamine activity by generalizing dopamine coding as always model-free versus always model-based. We never did that. Indeed, distinguishing model-based and model-free is generally very difficult even when a task is designed to do so. The logic is the other way around. Reviewers raised the point that dopamine activity following a primitive rule does not rule out dopamine's access to higher information. We totally agree. However, it is important to seek for a most parsimonious way to explain data, let alone a predominant theory or model that has long existed in the field. The significance of the present study is that we provide a parsimonious and unified explanation of dopamine signals in a task, with which some previous studies proposed more complex interpretations of dopamine signals. In the same logic, our results do not completely dispute the previous explanation. However, we believe that it does raise a question in the validity of the previous explanations.

To avoid reader's confusion, we added a new discussion in subsection “Confidence and TD errors”

As we emphasize in this section, our aim is not to dissociate model-based versus model-free function. We added a potential interaction in the last paragraph:

“Whereas our model takes a primitive strategy to estimate state value, state value can be also estimated with different methods. […] Together, our results and these models underscore the importance of considering moment-by-moment dynamics, and underlying computation.”

10) Please rephrase your interpretation as follows: " our modeling approach is consistent with activity at DA terminals functioning within the lines of a TD error framework in this particular reward-learning task, while we acknowledge that it is likely these signals are also acting outside of this framework in other tasks".

Thank you for the suggestion. We believe that in the previous manuscript, we clearly stated that there are some movement-related activity consistent with a previous study (Parker et al., 2016). This activity is clearly not a TD error, as analyzed in their subsequent study (Lee et al., 2019). In this sense, we did more than stating “these signals are also acting outside of this framework in other tasks”. Our data showed non-TD error signals in the present task.

Furthermore, we believe that “acting” is a misleading term because the present study primarily addresses the nature of signals or the information conveyed by these signals (i.e. the question of what is represented), rather than what they do (which may be implied by “acting”). It is our position to distinguish the question of representation (i.e. what information they convey) from functions (i.e. what they do). As we have discussed before (e.g. Kim et al., bioRxiv. 2019), TD errors can function as reinforcement as well as a regulator of motivation (e.g. if downstream neurons can integrate dopamine signals over time, RPE can be converted to value-like quantity) or a regulator of excitability of downstream neurons (as opposed to plasticity). We do not make any claim that dopamine signals do not function outside the traditional reinforcement learning framework.

Reviewers raised a question about generality of our findings. As we repeated, our main findings are three points.

1) Similarity of dopamine RPE signals in decision-task

2) Difference of baseline for the RPE

3) Correlates of confidence can naturally emerge in primitive TD error without additional variables "confidence"

We strengthened our discussion of each part to avoid reader's confusion. Related to the answers above, we emphasized the distinction between our conclusions and discussion, repeating our aims several times. We also added more results and discussion about movement-related activity.

11) It is crucial to report whether animals engage in licking in the water (or sample) port before the reward is provided? Does this covary with model 'value' and confidence? Does it covary with FP signals?

We agree on the importance of this point for two reasons. First, such measurements would confirm that mouse behaviors actually co-vary with confidence in our tasks. Second, more behavioral measurements can clarify if dopamine signals might be driving actions or vice-versa. Because we did not have a lick sensor in our freely-moving setup yet, we did not detect licks. To address these two questions, we first measured a leave frequency across different confidence levels to examine whether mice show behaviors modulated by confidence. In fact, we confirmed that a leave frequency was slightly, but significantly increased when mice made a choice error compared to when they made a correct choice, consistent with a previous report (Kepecs et al., 2008). Because the leave frequency was low overall, and we used a fixed water delay, we could not breakdown trial types further. We added this results in Figure 1H.

“Animals were required to stay in a water port for 1s to obtain water reward. However, in rare cases, they exited a water port early, within 1s after water port entry. We examined the effects of choice accuracy (correct or error) on the premature exit (Figure 1H). We found that while animals seldom exited a water port in correct trials, they occasionally exited prematurely in error trials, consistent with a previous study (Kepecs et al., 2008).”

“Since animals occasionally exit the water port prematurely in error trials (Figure 1H), we performed the same analyses as Figure 6C excluding trials where animals exited the water port prematurely. The results, however, did not change (Figure 7—figure supplement 5).”

To address potential movement signals, we also compared reaction time (odor port in ~ odor port out) or movement time (odor port out ~ water port in) and dopamine axon activity. First, reaction time was not correlated with dopamine axon signals. The result is shown in Figure 7—figure supplement 3. Second, while we found slight modulation of dopamine axon signals by movement time, movement time was not correlated with sensory evidence, and cannot explain dopamine axon activity patterns. We present this new result in Figure 7—figure supplement 4. Finally, to examine effects of any gross movement on dopamine axon signals, we performed video analyses using DeepLabCut, and examined relationship between GCaMP signals, body movement and control fluorescence (tdTom) signals. We found that body movement and tdTom signals cannot explain dopamine axon activity patterns. These results are shown in Figure 7—figure supplement 6.

[Editors' note: further revisions were suggested prior to acceptance, as described below.]

The manuscript has been improved but there are some remaining issues that need to be addressed before acceptance, as outlined below:In subsection “Similarity of dopamine axon signals across the striatum” paragraph three, optimism in Q learning should not be described without qualification. The fact that Q learning uses max_b Q(s(t+1),b) at the next state (s(t+1)) is not thought of being borne of optimism – this in fact reflects what the subject thinks it could do at that state (although it would indeed be optimistic to think that it would always do that). Confusion may arise in that optimistic Q values are associated with the motivation to explore. If the authors wish to include the assertion about Q learning, they should expand on their exact point.

Thank you for indicating the potential confusion that may arise from our phrasing. Although we did not explore action values in this study, higher criteria in Q-learning and in dopamine axon signals in DMS potentially suggest a common feature in learning such as function to push toward optimal actions (or state transitions). Since Q-learning and optimistic dopamine neurons have never been connected, we hoped that our data may encourage this new idea. At this moment, however, we totally agree with the reviewer’s suggestion, and we now explicitly mention that we have not found connection yet because the meaning of optimism may not be exactly the same.

“Optimistic expectation echoes with the idea of Watkin's Q-learning algorithm (Watkins, 1989; Watkins and Dayan, 1992) where an agent uses the maximum value among values of potential actions to compute RPEs, although we did not explore action values explicitly in this study. Future studies are needed to find the functional meaning of optimism in dopamine neurons and to examine whether the optimism is responsible for a specific learning strategy in DMS.”

The" dip" in the signal is important for the TD error. And if the proxy for the DLS signals is "more excitation" represented in Figure 8, this should still predict a dip in the DLS (albeit a smaller one). However, signals in DLS still show an increase in activity from baseline during water omission. This should be clarified.

As the reviewer points out, the difference in “baseline” can be regarded as a departure from the original TD errors. However, the overall dynamics are similar among the three regions of the striatum. Our conclusion that dopamine signals in all three areas is similar to TD errors is based on a more holistic approach, considering temporal dynamics associated with various events in a trial. Although the overall positive or negative reward responses of DLS and DMS dopamine axons, respectively are important deviations from TD errors, it would be important to point out similarity as well as the difference. We believe that our manuscript unambiguously demonstrates the difference (thus, TD errors as originally formulated). To clarify these points, we added the following paragraph in the first section of Discussion.

“The positively or negatively biased reward responses in DLS and DMS can be regarded as important departures from the original TD errors, as it was originally formulated (Sutton and Barto, 1998). […] In any case, the different baselines in TD error-like signals that we observed in instrumental behaviors can provide specific constraints on the behaviors learned through dopamine-mediated reinforcement in these striatal regions.”

We did not observe clear excitation or inhibition time-locked at water timing in error trials in dopamine axons in DLS. Whether they never show detectable inhibition with actual water omission is important to address in future.

“It is important to examine whether these dopamine neurons show detectable inhibition with large negative prediction error such as actual reward omission in an easy task, as the model predicts.”

The authors propose that the observed signals look like the scalar difference between the expected an observed value (with the exception of the DLS), but are not necessarily used to attach this scalar value product to the antecedent cue or action. This should be specifically and explicitly stated in the text as it is not necessarily scalar value that attaches to antecedent events, which is alluded to by referencing and modeling using the model-free TD algorithm.

Our main conclusion is that dopamine signals differ across different populations. Thus, our results clearly indicate that dopamine signals are not a scaler value, and add to the increasing literature indicating the heterogeneity of dopamine signals. It is, therefore, unclear why the reviewer thought that we are proposing a scalar dopamine signal. That is clearly not what we are proposing.

The idea that TD error equals a scalar signal has to be abandoned and this study is not the first to indicate that. It is also important to notice that there has been increasing appreciation that dopamine signals can vary while these signals are still consistent with TD errors. Therefore, the observation that dopamine signals are on average consistent with TD errors does not indicate that dopamine signals are a scaler value. Recent studies have started exploring the significance of variability in TD errors. That is, vector-type TD errors can provide computational advantages, for example, through distributional reinforcement learning (Dabney et al., 2020; Lowet et al., 2020). In these ideas, the important point is that the variation can exist within TD errors, and the variation in TD errors can garner certain computational powers. We believe that the present results indeed extend these ideas by showing that a type of variation can enable learning of different types of behavior.

Temporal difference error is defined by δ=r(St)+γ•V(St+1)−V(St). In this study, we used this definition to analyze dopamine signals. Our analyses, however, do not rely on a certain learning algorithm (either model-free or model-based). We have indeed avoided using a specific learning algorithm in order to avoid being too specific with respect to learning algorithms as we have indicated in the previous manuscript:

“To examine how the value and RPE may change within a trial, we employed a Monte-Carlo approach to simulate an animal’s choices assuming that the animal has already learned the task. We used a Monte-Carlo method to obtain the ground truth landscape of the state values over different task states, without assuming a specific learning algorithm.”

In other words, we are agnostic to how values (i.e. V(St+1),V(St)) and TD errors (δ) are computed.

To clarify these points, we have added the following paragraph. Although, in our view, a discussion on model-based and model-free are somewhat orthogonal to the present discussion, we now discuss this point, and cite three papers on this specific issue (Starkweather and Uchida; 2020; Landon et al., 2018; Akam and Walton, 2021):

“The positively or negatively biased reward responses in DLS and DMS can be regarded as important departures from the original TD errors, as it was originally formulated (Sutton and Barto, 1998). […] In any case, the different baselines in TD error-like signals that we observed in instrumental behaviors can provide specific constraints on the behaviors learned through dopamine-mediated reinforcement in these striatal regions.”

We also expanded future directions related to learning in Discussion.

“In this study, we modeled dynamical representation patterns of dopamine neurons in a steady state, but did not examine relationship between dopamine activity and actual learning. Especially, while our model used a discrete single stimulus state in each trial, it is naturalistic that animals use the experience from a single trial to update value in other stimulus states (Bromberg-Martin et al., 2010), and/or animals represent states in a more continuous manner (Kiani and Shadlen, 2009). It is important in the future to examine how dynamical and diverse dopamine signals are used during learning and/or performance.”

Please include summary statistics in additional to P values where appropriate (e.g., for ANOVA and t-tests).

We added a supplementary source file in each figure to show summary statistics. We noticed that we had used dF/F for some of statistics in previous versions. To be consistent throughout the paper, now we use z-score in all the statistics and clearly state it in the summary statistics.